# Inter-species population dynamics enhance microbial horizontal gene transfer and spread of antibiotic resistance

Robert M Cooper[1]*, Lev Tsimring[1,2], Jeff Hasty[1,2,3,4]*

[1]BioCircuits Institute, University of California, San Diego, San Diego, United States; [2]San Diego Center for Systems Biology, University of California, San Diego, San Diego, United States; [3]Molecular Biology Section, Division of Biological Science, University of California, San Diego, San Diego, United States; [4]Department of Bioengineering, University of California, San Diego, San Diego, United States

**Abstract** Horizontal gene transfer (HGT) plays a major role in the spread of antibiotic resistance. Of particular concern are *Acinetobacter baumannii* bacteria, which recently emerged as global pathogens, with nosocomial mortality rates reaching 19–54% (Centers for Disease Control and Prevention, 2013; Joly Guillou, 2005; Talbot et al., 2006). *Acinetobacter* gains antibiotic resistance remarkably rapidly (Antunes et al., 2014; Joly Guillou, 2005), with multi drug-resistance (MDR) rates exceeding 60% (Antunes et al., 2014; Centers for Disease Control and Prevention, 2013). Despite growing concern (Centers for Disease Control and Prevention, 2013; Talbot et al., 2006), the mechanisms underlying this extensive HGT remain poorly understood (Adams et al., 2008; Fournier et al., 2006; Imperi et al., 2011; Ramirez et al., 2010; Wilharm et al., 2013). Here, we show bacterial predation by *Acinetobacter baylyi* increases cross-species HGT by orders of magnitude, and we observe predator cells functionally acquiring adaptive resistance genes from adjacent prey. We then develop a population-dynamic model quantifying killing and HGT on solid surfaces. We show DNA released via cell lysis is readily available for HGT and may be partially protected from the environment, describe the effects of cell density, and evaluate potential environmental inhibitors. These findings establish a framework for understanding, quantifying, and combating HGT within the microbiome and the emergence of MDR super-bugs.

DOI: https://doi.org/10.7554/eLife.25950.001

*For correspondence:
rcooper@ucsd.edu (RMC);
jhasty@eng.ucsd.edu (JH)

**Competing interests:** The authors declare that no competing interests exist.

## Introduction

The spread of antibiotic resistance among pathogenic microbes is a major and growing threat to public health. Gram-negative *Acinetobacter* spp. are a particularly worrisome example - these bacteria thrive in hospital settings, causing around 9% of nosocomial infections, particularly in the respiratory tract (*Joly-Guillou, 2005*). This prevalence, combined with high levels of antibiotic resistance (*Antunes et al., 2014*), has led the Infectious Diseases Society of America to designate *Acinetobacter baumanii* one of six particularly problematic multidrug-resistant (MDR) pathogens (*Talbot et al., 2006*) and the US Centers for Disease Control to assign it threat level 'Serious' (*Centers for Disease Control and Prevention (CDC), 2013*).

Much of the threat posed by *Acinetobacter* stems from its ability to acquire drug resistance via horizontal gene transfer (HGT) (*Peleg et al., 2012*). *Acinetobacter* has a remarkably high rate of HGT (*Touchon et al., 2014*), and many antibiotic resistance genes in clinical isolates appear to have been recently acquired from other human pathogens (*Adams et al., 2008*; *Fournier et al., 2006*;

**eLife digest** Every year, antibiotics save millions of lives, but this may not last forever. The bacteria that cause infections are getting smarter and continuously evolve genes to become resistant to antibiotics, which makes it harder to kill them. In many cases, using stronger drugs can bypass this problem, but some 'super-bugs' are developing resistance to every drug we have. For example, the bacterium *Acinetobacter baumannii* has recently been classified as a global threat that kills thousands of people every year, which placed it on a top six 'most wanted' list for multi drug-resistant bacteria. Worryingly, this drug resistance seems to develop faster than 'standard' evolution would allow, making it difficult to keep up with developing new effective drugs.

One way bacteria can shortcut the evolution of resistance is through a process called horizontal gene transfer, in which they collect resistance genes from other bacteria. Some bacteria can speed up this gene transfer by actively killing their neighbors to extract their DNA. However, until now, this process has not been observed directly, and it was not fully understood where and when killing neighbors becomes important for gene transfer.

Now, Cooper, Tsimring and Hasty have studied a relative of *A. baumannii* called *A. baylyi*. Together with another type of bacteria that contained green fluorescence genes, *A. baylyi* was placed onto a surface that allowed both species to grow. As the two types of bacteria grew together, *A. baylyi* started to kill the other one and stole their genes. This happened so often that some started to become fluorescent, which could be observed in real time under a microscope. *A. baylyi* also stole genes for antibiotic resistance, and when an antibiotic was added, the bacteria with the stolen resistance genes kept growing and dividing, while the others were killed.

Cooper et al. then developed a mathematical model to quantify and simulate this killing-enhanced horizontal gene transfer. The results showed that killing other bacteria made gene transfer more effective when the number of *A. baylyi* was high and the number of 'victims' was low – and also when they were together for a shorter period.

This work may help to explain how *Acinetobacter* and similar bacteria develop drug resistance so quickly. A next step will be to measure and compare gene transfer parameters in different types of bacteria. A better understanding of how, where, and when gene transfer happens, may in the future help to guide strategies to fight resistance.

DOI: https://doi.org/10.7554/eLife.25950.002

*Imperi et al., 2011*). Although *A. baumanii* can be naturally competent in vitro under some conditions (*Ramirez et al., 2010*; *Wilharm et al., 2013*), it is not understood at the population level how *Acinetobacter* acquires foreign genes at such high rates in real-world conditions, particularly considering that extracellular DNA is rapidly degraded or sequestered (*Nielsen et al., 2007*). For example, DNA lost its transforming ability with a half life of around 1 hr in soil (*Nielsen et al., 2000*) and 1 min in saliva (*Mercer et al., 1999*), and ingested DNA was unable to transform even highly competent *A. baylyi* within the mouse gut (*Nordgård et al., 2007*).

Many *Streptococcus* species also actively exchange genetic material in natural environments. Some of these species use quorum sensing to enhance HGT by co-regulating competence (the ability to take up extracellular DNA) with secretion of diffusible bacteriocins (small protein toxins) (*Steinmoen et al., 2002*). These toxins cause the release of potential donor DNA by lysing nearby sister cells (fratricide) or closely related species (sobrinocide) (*Johnsborg et al., 2008*; *Kreth et al., 2005*; *Wei and Håvarstein, 2012*), thereby making their genes available for uptake. However, fratricide-enhanced HGT has not been observed outside *Streptococcus*, and the narrow target range of bacteriocins limits potential DNA donors. Another killing mechanism, the contact-dependent type-VI secretion system (T6SS), is conserved across a much broader range of bacteria (*Schwarz et al., 2010a*). Recently, *Vibrio cholera* was shown to similarly co-regulate T6SS expression with competence, also enhancing HGT within biofilms (*Borgeaud et al., 2015*).

In this work, we extend the observation of killing-enhanced HGT to *Acinetobacter*, and we develop an experimental and modeling framework to quantify the population dynamics of this phenomenon within microbial communities. As a model system, we used *Acinetobacter baylyi*, which is closely related to *A. baumannii* (*Touchon et al., 2014*), because it is genetically tractable, fully

sequenced, and well-studied (*Elliott and Neidle, 2011*). The two also share the key features of T6SS-mediated killing (*Weber et al., 2013*; *Carruthers et al., 2013*) and natural competence (*Ramirez et al., 2010*; *Wilharm et al., 2013*), and they are similar enough that standard phenotypic assays used in the clinic often fail to distinguish between them (*Chen et al., 2007*).

We show that *A. baylyi* uses its T6SS to lyse and acquire genes from neighboring *E. coli* cells, and that this process is frequent enough to observe functional and adaptive HGT from prey to predator cells in real time. Prey cell DNA released by predator cells is also available for uptake by other nearby, non-killing cells, highlighting the importance of polymicrobial population dynamics in HGT and the emergence of MDR. To understand the population dynamics of killing-enhanced HGT, we develop a model quantifying both neighbor killing and natural transformation. HGT parameters fit using naked DNA predict transfer of genes released via neighbor killing surprisingly well, suggesting DNA released from cell lysis in situ is readily available for uptake. Using this model, we characterize the impact of neighbor killing on HGT in a wide range of environmental conditions and evaluate potential inhibition strategies, and we experimentally confirm the key predictions. Contact-dependent neighbor killing may be a widespread contributor to HGT among gram-negative bacteria, and for *Acinetobacter* in particular, killing-enhanced HGT may play a key role in the emergence of clinically pervasive MDR 'super-bug' strains. In this context, our dynamic model should be useful for predicting and combating HGT of antibiotic resistance.

## Results

### Killing-enhanced horizontal gene transfer in *Acinetobacter*

To study the population dynamics generated by contact-dependent cell killing, we designed a custom microfluidic device (*Figure 1—figure supplement 1*) enabling visualization of spatially structured communities at single-cell resolution in a monolayer. We seeded the device with a simplified community consisting of predator *A. baylyi* strain ADP1 (*Elliott and Neidle, 2011*) (the predator) that kill directly adjacent, GFP-expressing *Escherichia coli* (the prey) (*Basler et al., 2013*). As expected, we observed spontaneous lysis of individual *E. coli* adjacent to *Acinetobacter* cells. Serendipitously, a few *Acinetobacter* cells spontaneously began expressing GFP, suggesting that they had acquired the GFP gene via HGT from DNA released by lysed *E. coli*.

In our initial experiments, the GFP-expressing plasmid in *E. coli* had a ColE1 origin of replication that does not stably replicate in *Acinetobacter* (*Palmen et al., 1993*). To better observe HGT, we constructed the broad-host plasmid pBAV1k-GFP (*Bryksin and Matsumura, 2010*), which can replicate in both *Acinetobacter* and *E. coli*. Expression of the GFP gene from this plasmid is repressed in *E. coli* by LacI, which is absent in *Acinetobacter*. We transformed *E. coli* with this plasmid and then seeded them in our microfluidic device alongside mCherry-expressing *Acinetobacter* (*Figure 1a–d*, *Videos 1–6*). After 3 hr, *Acinetobacter* had lysed a large number of *E. coli* cells, and multiple independent HGT events were visible within the device's approximately $10^4$ um$^2$ traps (*Figure 1b–d*). Importantly, horizontally acquired GFP was stably maintained during cell division, giving rise to clumps of cells that were both red and green. In the movies, note that although GFP expression in *E. coli* was repressed by LacI, some *E. coli* appeared green due to incomplete repression (see also Supplementary Note 1).

HGT of antibiotic resistance allows pathogenic bacteria to survive treatment with antibiotics that would kill the parental strain. In our experiments, the transferring plasmid, pBAV1k-GFP, contained a kanamycin resistance gene in addition to GFP. Therefore, expression of GFP should indicate a newly kanamycin-resistant strain of *Acinetobacter*, whose de novo appearance would demonstrate the potential clinical relevance of HGT within a microbial community. To test whether this directly observed HGT would be enough to provide a population-level selective advantage, we used our microfluidic chip to visualize functional HGT of antibiotic resistance in real time. We seeded the microfluidic device with mCherry-expressing *Acinetobacter* alongside *E. coli* carrying pBAV1k-GFP, grew them for just under 12 hr to allow *E. coli* lysis and HGT to occur, and then added kanamycin (*Videos 7,S8*). Within about 7 hr of kanamycin addition, multiple HGT events could be visibly identified by the emergence of GFP-expressing *Acinetobacter* (arrows in *Figure 1e–g*). Only those *Acinetobacter* that were expressing GFP, indicating horizontal transfer of kanamycin resistance, continued to grow (*Figure 1h*, *Videos 7,8*), while the parental GFP-negative *Acinetobacter* cells became

Images corresponding to **Video 3**

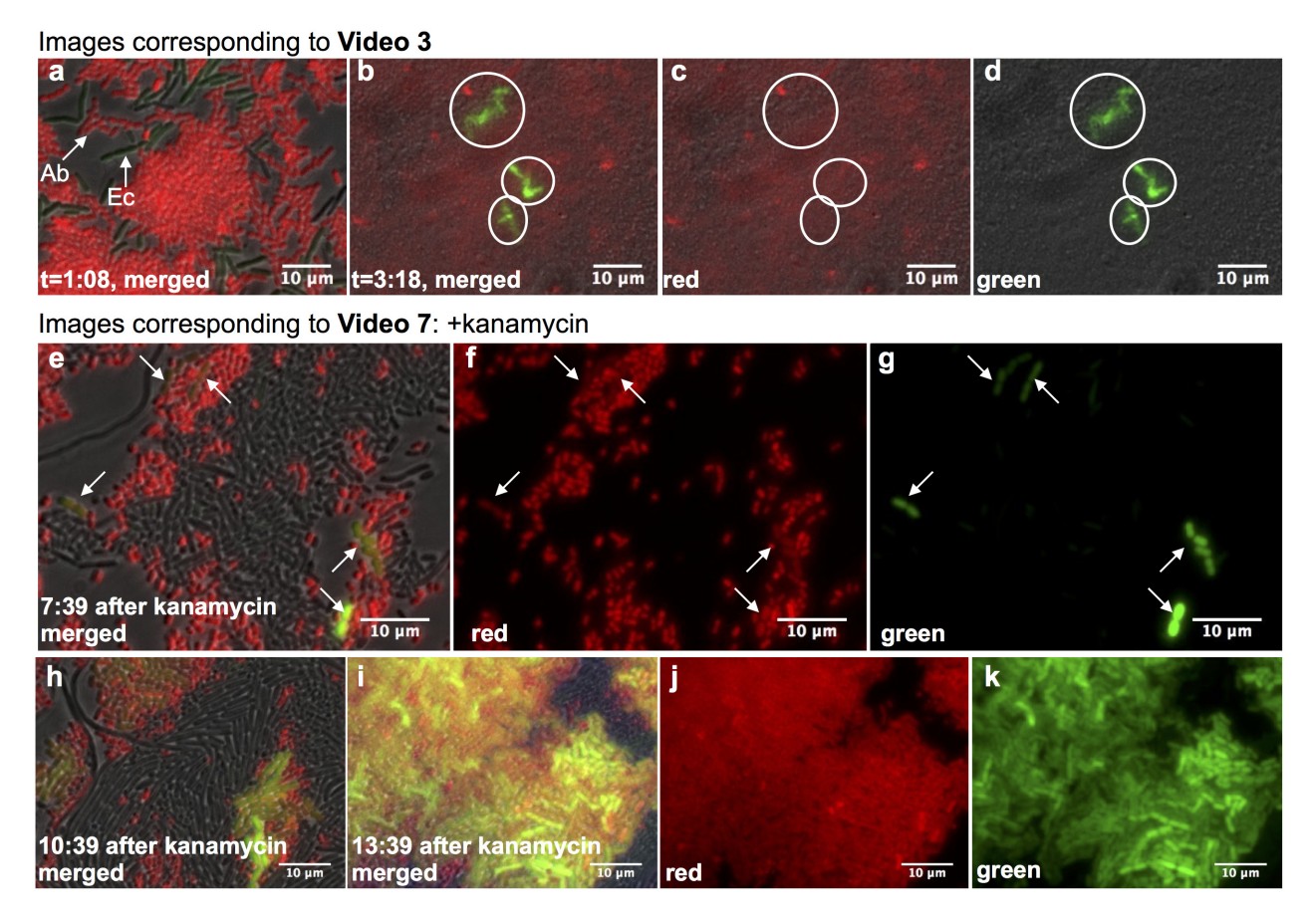

Images corresponding to **Video 7**: +kanamycin

**Figure 1.** Real-time observation of functional HGT in microfluidic traps. (**a–d**) *Acinetobacter* expressing mCherry were mixed with *E. coli* carrying LacI-repressed pBAV1k-GFP inside a microfluidic device (frames captured from *Video 3*, see also Supplementary Note 1). (**a**) t = 1:08, both *Acinetobacter* (Ab) and *E. coli* (Ec) were present, and no HGT had occurred. (**b–d**) t = 3:18, all *E. coli* in the field had been lysed, and several independent lineages of GFP-expressing *Acinetobacter* deriving from HGT were visible (circled). (**e–k**) HGT renders *Acinetobacter* resistant to antibiotic treatment (frames captured from *Video 7*). Kanamycin was added 11:42 after seeding the device. Still images were captured at indicated times after kanamycin addition, after several independent HGT events had already occurred (arrows in e-g). (**h–k**) Newly kanamycin-resistant *Acinetobacter* began outcompeting both *E. coli* and the sensitive parent (**h**), and by 13 hr after kanamyin addition (**i–k**), they dominated the environment.

DOI: https://doi.org/10.7554/eLife.25950.003

The following figure supplement is available for figure 1:

**Figure supplement 1.** Microfluidic chip design.

DOI: https://doi.org/10.7554/eLife.25950.004

smaller and stopped dividing. The red and green, dual-labeled *Acinetobacter* quickly dominated the device, lysing neighboring *E. coli* (dark) and pushing the non-dividing, GFP-negative *Acinetobacter* cells out of the trap (*Figure 1i–k*). To our knowledge, these experiments represent the first real-time observation of adaptive, cross-species HGT via natural competence that rapidly enables invading cells to thrive in a new niche.

The type VI secretion system (T6SS) of *Vibrio cholera* was recently shown to enhance HGT by promoting the release of DNA from lysed prey cells, thereby making it available for uptake (*Borgeaud et al., 2015*). We hypothesized that T6SS-mediated killing was similarly contributing to the high rate of HGT we observed with *Acinetobacter*. To test this, we quantified HGT from *E. coli* to *Acinetobacter* using bulk experiments, in which we dried small spots of liquid culture onto agar plates. These experiments involve surface-attached cells that are developing spatially-structured communities. This is a qualitatively different condition from growth in shaking culture tubes and

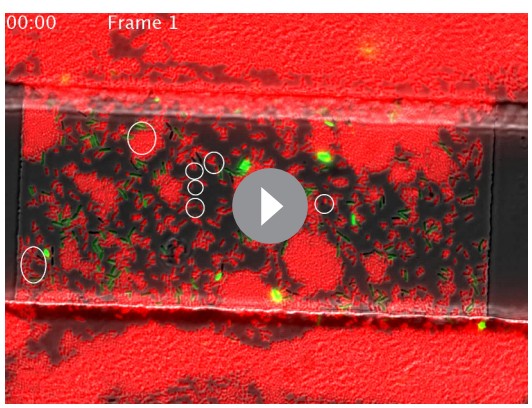

**Video 1.** Real-time observation of HGT from *E. coli* to *Acinetobacter*. *E. coli* carrying pBAV1k-PLac-GFP were mixed with *Acinetobacter* expressing mCherry in a microfluidic device. Most *E. coli* were rapidly killed, and multiple HGT events were observed across several traps (circled). PLac-GFP was not fully repressed within *E. coli*, despite the presence of LacI (not found in *Acinetobacter*), so some *E. coli* expressed GFP as well. Expression of mCherry in *Acinetobacter* faded toward the end of the movies (see also Supplemental Note 1). Individual movies are physically separated traps within the same microfluidic chip.

DOI: https://doi.org/10.7554/eLife.25950.005

more similar to biofilm dynamics, although we did not study true biofilms, which take longer to mature.

In these bulk experiments, we co-cultured *Acinetobacter* carrying genomic spectinomycin (spect) resistance and *E. coli* carrying both a genomic chloramphenicol (cm) resistance marker and the 'bait' kanamycin (kan) resistance plasmid pBAV1k. We placed the cm resistance gene in a region of the *E. coli* genome with minimal homology to *Acinetobacter* to preclude its horizontal transfer, enabling us to quantify the remaining *E. coli* and thereby measure killing efficiency.

We co-cultured *Acinetobacter* (spect) and *E. coli* (cm+kan), seeded together at roughly equal concentrations, on agar plates at 30°C overnight to allow for T6SS-mediated killing and HGT. We also cultured each species alone as controls. We quantified the efficiency of *E. coli* lysis and HGT of kanamycin resistance by resuspending the initial spots, spotting serial dilutions of that resuspension onto selective agar plates (spect, cm +kan, and spect+kan), and counting colony-forming units (CFUs). As expected, *Acinetobacter* dramatically reduced *E. coli* numbers during co-culture (*Figure 2a*, cm+kan). Simultaneously, a novel spect+kan double antibiotic-resistant phenotype appeared. We picked three clones with this novel spect+kan phenotype and confirmed them to be *Acinetobacter* by microscopy and 16S sequencing. Transfer of the genomic cm *E. coli* marker to *Acinetobacter* was below our limit of detection (2500 CFUs in this case), as we did not recover any spect+cm resistant cells.

Next, we confirmed the role of T6SS-mediated killing in HGT to *Acinetobacter* by replacing the T6SS structural gene *hcp* (*Carruthers et al., 2013*; *Basler et al., 2013*) with tetracycline resistance. To compare these non-killing *Acinetobacter* Δ*hcp* with the wild type (WT), we incubated *E. coli* carrying pBAV1k on agar plates at 37°C for 150 min either alone or mixed with WT, Δ*hcp*, or both strains of *Acinetobacter*, with *Acinetobacter* at optical

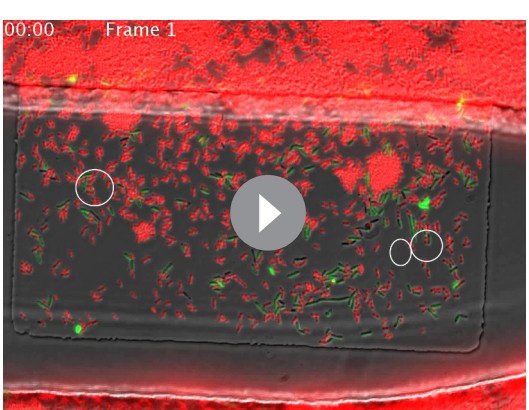

**Video 2.** Real-time observation of HGT from *E. coli* to *Acinetobacter*. See caption to *Video 1* for details. Individual movies are physically separated traps within the same microfluidic chip.

DOI: https://doi.org/10.7554/eLife.25950.006

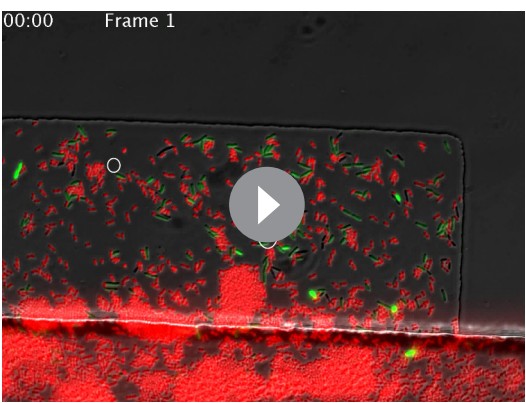

**Video 3.** Real-time observation of HGT from *E. coli* to *Acinetobacter*.
See caption to *Video 1* for details. Individual movies are physically separated traps within the same microfluidic chip.

DOI: https://doi.org/10.7554/eLife.25950.007

density (OD) five and *E. coli* at OD 1 (*Figure 2b, c*). No *E. coli* were detected after co-incubation with WT *Acinetobacter* (*Figure 2b*, second bar). The limit of detection was 10 CFUs, which was significantly different from all other conditions at $p = 0.011$ (see Materials and methods for statistical tests). In contrast, survival of *E. coli* mixed with *Acinetobacter* Δ*hcp* was not different from cultures containing *E. coli* alone ($p = 0.063$), confirming a dramatic reduction of killing efficiency for *Acinetobacter* Δ*hcp* (*Figure 2b*, third bar). At the same time, elimination of killing impaired HGT by about 100-fold (*Figure 2c*, dark bars, $p<0.001$). Impaired HGT was not due to a competence defect, since WT and Δ*hcp* *Acinetobacter* had equal competence for purified plasmid DNA (*Figure 2d*, $p = 0.57$). Furthermore, in a time course experiment with each *Acinetobacter* strain co-cultured at 37°C with approximately equal concentrations of *E. coli* (*Figure 2e–g*), we detected the spect+kan dual

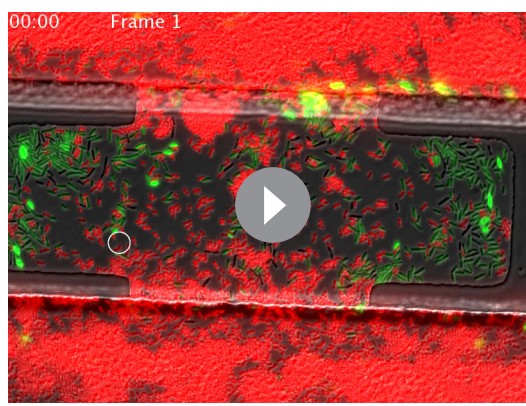

**Video 4.** Real-time observation of HGT from *E. coli* to *Acinetobacter*.
See caption to *Video 1* for details. Individual movies are physically separated traps within the same microfluidic chip.
DOI: https://doi.org/10.7554/eLife.25950.008

drug-resistant phenotype (indicating HGT) 2 hr earlier for WT cells (*Figure 2e*) than for non-killing Δ*hcp* cells (*Figure 2f*, also compare solid to dashed lines in *Figure 2g* for fraction of transformed *Acinetobacter* cells). Presumably, the residual HGT to *Acinetobacter* Δ*hcp* results from slow, but continued, spontaneous DNA release from *E. coli*, possibly from spontaneous cell lysis.

Together, these results and those reported for *Vibrio cholera* (*Borgeaud et al., 2015*) support a model in which the T6SS of *Acinetobacter* promotes HGT by killing neighboring *E. coli*, which then releases its DNA for subsequent uptake by *Acinetobacter*. If this model is correct, then WT *Acinetobacter* would complement HGT to the killing-defective Δ*hcp* *Acinetobacter* strain in trans by lysing neighboring *E. coli*, thereby releasing their DNA into an extracellular pool accessible to all competent cells. To test this, we mixed the tetracycline (tet)-resistant *Acinetobacter* Δ*hcp* with the spect-resistant WT *Acinetobacter* and co-cultured them in a '3-strain' spot along with *E. coli*, seeding *Acinetobacter* at total OD = 5 and *E. coli* at OD = 1. Survival of *E. coli* in the three-strain culture was reduced by about $10^4$-fold (*Figure 2b* fourth bar, $p<0.001$), although their survival was improved relative to *E. coli* mixed with only WT *Acinetobacter*, where *E. coli* were reduced below the 10 CFUs

limit of detection ($p = 0.011$), presumably due to the lower fraction of killing cells in the three-strain culture.

Consistent with the model, killing of *E. coli* by WT *Acinetobacter* in three-strain cultures was sufficient to increase HGT to co-cultured *Acinetobacter* Δ*hcp* by about 100-fold (*Figure 2c*, compare two- to three-strain communities in right bar group, $p<0.001$), while HGT to the WT remained unchanged relative to a two-strain community (*Figure 2c*, left bar group, $p = 0.68$). We found similar results for a time course experiment comparing HGT in two-strain (*Figure 2h*) and three-strain (*Figure 2i*) communities: co-culture with WT *Acinetobacter* dramatically increased HGT from *E. coli* to non-killing *Acinetobaceter* Δ*hcp*. Interestingly, even in three-strain communities, HGT to *Acinetobacter* Δ*hcp* was slightly less than HGT to the WT (*Figure 2c* light bars, $p<0.014$, and *Figure 2i*). This likely indicates a spatial effect -

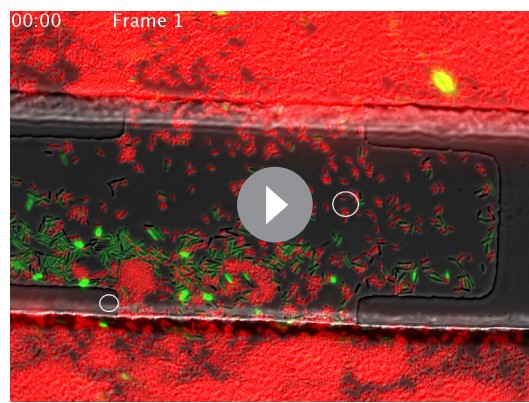

**Video 5.** Real-time observation of HGT from *E. coli* to *Acinetobacter*.
See caption to *Video 1* for details. Individual movies are physically separated traps within the same microfluidic chip.
DOI: https://doi.org/10.7554/eLife.25950.009

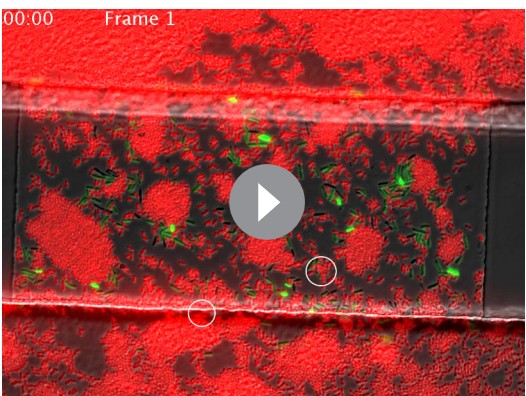

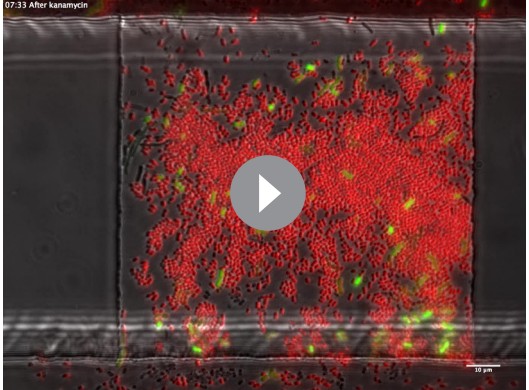

**Video 6.** Real-time observation of HGT from *E. coli* to *Acinetobacter*.
See caption to *Video 1* for details. Individual movies are physically separated traps within the same microfluidic chip.
DOI: https://doi.org/10.7554/eLife.25950.010

**Video 7.** Killing-enhanced HGT enables *Acinetobacter* to dynamically adapt to new environments. *Acinetobacter* and *E. coli* were grown together in a microfluidic chip as in *Videos 1–6* for several hours, after which the media was switched to include kanamycin, resistance to which is encoded on the transferring plasmid pBAV1k. The movies begin 7.5 hr after kanamycin addition. Only *Acinetobacter* expressing GFP continue to grow. These mCherry+/ GFP + cells push their mCherry+/GFP- parents out of the trap and kill neighboring *E. coli*.
DOI: https://doi.org/10.7554/eLife.25950.011

actively killing predator cells will be physically closest to the DNA released from their lysed prey. Together with experiments below showing HGT inhibition by extracellular DNase (Figure 7e), these results support a model in which the T6SS releases DNA into the extracellular environment via lysis of neighboring cells, thereby making it available for uptake by any nearby cell, but the T6SS is not directly involved in DNA uptake (*Figure 2j*).

To further characterize killing-enhanced HGT in *Acinetobacter*, we analyzed the effect of genetic context. Previous experiments have shown that *A. baylyi* is competent to take up and replicate plasmids with a limited set of origins of replication, they require genomic homology for efficient recombination into their genome, and both genomic and plasmid DNA can serve as a source for recombination (*Palmen et al., 1993*). However, those experiments were done using chemically purified, exogenously added DNA. Therefore, we experimentally confirmed them for the specific case of killing-enhanced HGT.

We co-cultured *Acinetobacter* on agar plates with *E. coli* containing each of three different bait plasmids conferring kan resistance: pBAV1k that replicates in both species, pRC03 that cannot replicate in *Acinetobacter*, and pRC03H - a derivative of pRC03 containing 3 kb of *Acinetobacter* genomic homology to promote genomic integration (*de Vries and Wackernagel, 2002*). *Acinetobacter* acquired the kan resistance gene from either pBAV1k or pRC03H, but not from pRC03 (*Figure 2—figure supplement 1a*). We also tested whether contact-dependent killing enables *Acinetobacter* to efficiently acquire genes from the *E. coli* genome. To do so, we created a bait plasmid containing kan resistance adjacent to 22 kb of *Acinetobacter* genomic homology. We transformed this plasmid into *E. coli* to create a plasmid bait strain, and then we integrated it into the *E. coli* genome to create a genomic bait strain (see Materials and methods).

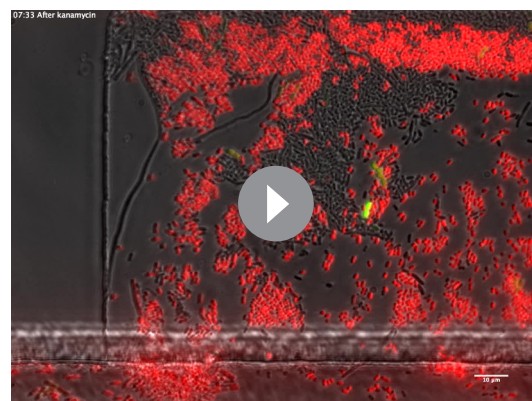

**Video 8.** Killing-enhanced HGT enables *Acinetobacter* to dynamically adapt to new environments. See caption to *Video 7* for details. Individual movies are physically separated traps within the same microfluidic chip.
DOI: https://doi.org/10.7554/eLife.25950.012

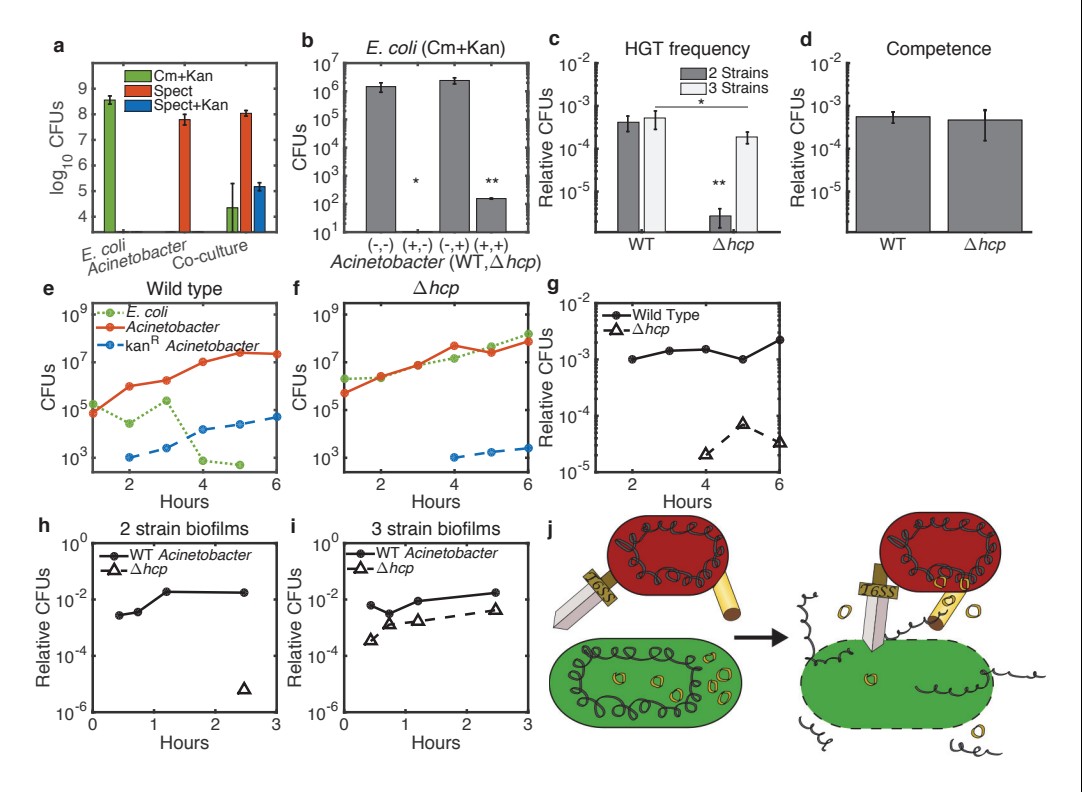

**Figure 2.** Enhancement of HGT efficiency by neighbor killing. Communities were seeded with 2 ul droplets containing indicated strains of *Acinetobacter* with genomic spect resistance, mixed with genomically cm-resistant *E. coli* carrying the kan-resistant donor plasmid pBAV1k. Spots were incubated at 30°C overnight (**a**) or at 37°C for 150 min (**b–d**) or for the indicated time (**e–i**). Lower limits on y-axes are the limits of detection. (**a**) *E. coli* (cm+kan, green bars), *Acinetobacter* (spect, orange bars), and newly double-resistant *Acinetobacter* (spect+kan, blue bars) present after growth either alone or in co-culture. (**b**) Survival of *E. coli* after growth alone (-,-) or with wild type (+,-), non-killing Δ*hcp* (-,+), or both (+,+) strains of *Acinetobacter*. Spots were seeded with *Acinetobacter* at optical density (OD) five and *E. coli* at OD 1. (**c**) HGT during the experiment shown in b, measured as the proportion of wild type (WT - spect, left bar group) or Δ*hcp* (tetracycline, right bar group) CFUs that were also resistant to kan. Two-strain cultures (dark bars) correspond to the center two bars in b, while the three-strain cultures (light bars) corresponds to the rightmost bar in b. (**d**) Transformation efficiency of *Acinetobacter* as in c, but mixed with purified pBAV1k DNA just before spotting. (**e–g**) Time course showing CFUs of *E. coli* (dotted green), total *Acinetobacter* (solid orange) and double-resistant *Acinetobacter* (dashed blue) in two-strain cultures (wild-type *Acinetobacter* in e and Δ*hcp* *Acinetobacter* in f). (**g**) The proportion of wild type (solid line) and Δ*hcp* (dashed line) *Acinetobacter* cells that have acquired kan resistance during the experiment in e-f. (**h,i**) Time course showing HGT complementation by killing in trans. The proportion of double-resistant cells is shown for wild type (solid lines) and Δ*hcp* (dashed lines) *Acinetobacter* grown with *E. coli* either separately in two-strain (**h**) or together in three-strain (**i**) cultures. (**j**) A conceptual model for killing-enhanced HGT to *Acinetobacter*. Statistical significance levels are: ∗ = $p<0.05$ with significance calculated using raw data, and ∗∗ = $p<0.001$, with significance calculated on log10-transformed data (see Materials and methods for sample sizes and statistical analysis, and see main text for exact p-values).

DOI: https://doi.org/10.7554/eLife.25950.013

The following figure supplements are available for figure 2:

**Figure supplement 1.** Effect of genomic context on killing-enhanced HGT.

DOI: https://doi.org/10.7554/eLife.25950.014

**Figure supplement 2.** Confirmation of HGT from replicating and homology plasmids.

DOI: https://doi.org/10.7554/eLife.25950.015

*Acinetobacter* acquired the kan resistance gene regardless of its location (*Figure 2—figure supplement 1b*), but HGT was approximately 100-fold lower when the kan gene was in the *E. coli* genome. Considering that the bait plasmid had a pBR322 origin of replication, with 10–20 plasmids per cell, the per-copy difference was only about 5- to 10-fold.

Finally, we picked three double-resistant *Acinetobacter* clones each that had acquired either the replicating or homology plasmid for further analysis, isolated both plasmid and genomic DNA, and confirmed that the replicating plasmid pBAV1k had transferred as an episomal unit, whereas the

# Box 1.

### Species

$A_1 = $ Parent *Acinetobacter*
$A_2 = $ Transformed *Acinetobacter*
$A_T = A_1 + A_2$
$E = E.coli$
$D = $ Plasmid DNA
$D_G = $ Genomic DNA
$N^* = $ Number of cells ($A$ or $E$) on the perimeter of micro $-$ colonies
$N_0 = $ Initial number of cells ($A$ or $E$)

### Parameters

$\gamma_{A,E} = $ Growth rates
$K_{A,E} = $ Growth saturation
$r_{kill} = $ Killing rate
$K_{kill} = $ Killing saturation
$K_{E\_kill} = $ Prey killing saturation factor
$c = $ DNA uptake rate
$K_{DNA} = $ DNA saturation
$\epsilon = $ HGT efficiency
$p = $ Plasmids per cell
$G = D_G$ per cell
$r_{leak} = $ Spontaneous DNA release rate

### Reactions

$D + A\_19 \xrightarrow{c\epsilon} A_2$
$D + A_1 \xrightarrow{c(1-\epsilon)} A_1$
$D + A_2 \xrightarrow{c} A_2$
$D_G + A_T \xrightarrow{c} A_T$
$A_T + E \xrightarrow{r_{kill}} A_T + pD + G$
$E \xrightarrow{r_{leak}} E + pD + G$

### Equations

$$\left. \begin{aligned} \frac{dA_1}{dt} &= \gamma_A A_1 - c\epsilon \frac{A_1 D}{K_{DNA}+D+D_G} \\ \frac{dA_2}{dt} &= \gamma_A A_2 + c\epsilon \frac{A_1 D}{K_{DNA}+D+D_G} \\ \frac{dE}{dt} &= \gamma_E E - r_{kill} \frac{A_T^* E^*}{K_{kill}+A_T^*+K_{E\_kill}E^*} \\ \frac{dD}{dt} &= -c\frac{A_T D}{K_{DNA}+D+D_G} + p\left(r_{kill}\frac{A_T^* E^*}{K_{kill}+A_T^*+K_{E\_kill}E^*}+r_{leak}E\right) \\ \frac{dD_G}{dt} &= -c\frac{A_T D_G}{K_{DNA}+D+D_G} + G\left(r_{kill}\frac{A_T^* E^*}{K_{kill}+A_T^*+K_{E\_kill}E^*}+r_{leak}E\right) \end{aligned} \right\} \left(1-\frac{A_T}{K_A}\right)\left(1-\frac{E}{K_E}\right)$$

$$N^* = N\left(\frac{1}{1+\left(\frac{N}{5N_0}\right)^2}\right) + \pi N_0\left(\sqrt{\frac{N}{N_0}}-1\right)\left(\frac{\left(\frac{N}{5N_0}\right)^2}{1+\left(\frac{N}{5N_0}\right)^2}\right)$$

DOI: https://doi.org/10.7554/eLife.25950.016

homology plasmid pRC03H had integrated into the genome (*Figure 2—figure supplement 2*). These results do not rule out additional lower frequency mechanisms, but they indicate that previous characterizations of competence using purified DNA are equally applicable to in situ HGT within spatially structured microbial communities.

## Quantifying microbial population dynamics of killing-enhanced gene transfer

While killing-enhanced HGT has now been observed in multiple genera, little is known about the population dynamics of this process, its efficiency, or how that efficiency is influenced by environmental conditions. This lack of understanding is largely due to the absence of a quantitative method to measure mechanistic HGT parameters. Currently, HGT is generally quantified using the fraction of transformed CFUs or transformed CFUs per ng of DNA. This is problematic, because results depend on multiple parameters extrinsic to inherent competence, including cell concentration, DNA concentration, and incubation time. This makes results difficult to compare across experiments, conditions, strains, or even labs. A more useful approach would be to quantify transformation using variables intrinsic to the cells and the DNA, such as the DNA uptake rate and transformation efficiency per molecule of DNA. To this end, we developed a population dynamic model of spatially structured microbial communities that couples transformation via natural competence and contact-dependent killing (summarized in *Box 1* and explained in more detail in the Materials and methods).

The model parameters pertaining to growth, transformation, and killing can each be measured sequentially in simplified conditions (*Figure 3*, *Figure 3—figure supplements 1–3*, *Table 1*). For all parameter fitting, we incubated cells in spots on agar plates, the same condition used to measure HGT in dual-species communities (see Materials and methods). First, we measured growth parameters for each species separately (*Figure 3—figure supplements 1–2*, and see Materials and methods). Second, we fit the natural transformation parameters using data obtained by mixing *Acinetobacter* with known concentrations of the genomically integrating kan resistance plasmid pRC03H-2S, derived from pRC03H used earlier, with higher efficiency integration via fully homolgous recombination (HR) (*Palmen et al., 1993*), rather than homology-facilitated illegitimate recombination (*de Vries and Wackernagel, 2002*) as in *Figure 2*, see Materials and methods. We incubated the cell-DNA mixtures on agar plates, and then we counted the number of cells that had acquired kan resistance. To obtain values for the HGT parameters, we simultaneously fit data from time courses with either limiting (*Figure 3a*) or saturating (*Figure 3b*) DNA, and from DNA dilution series with cells harvested at different time points (*Figure 3c*). Third, we fit parameters for T6SS-mediated killing by mixing different concentrations of *E. coli* and *Acinetobacter*, incubating them together on agar plates, and counting the cells of each type that were present upon harvesting. As with DNA uptake, we fit killing parameters simultaneously for several time courses (*Figure 3d–i* and *Figure 3—figure supplement 3*) and dilution series (*Figure 3j*). Finally, to determine the 'leak' rate of DNA from *E. coli*, we measured HGT to non-killing *Acinetobacter* Δhcp and found the best fit for $r_{leak}$, with the killing rate $r_{kill}$ set to 0 in the model (*Figure 3k,l*). The leak rate may be due to spontaneous *E. coli* cell death, but we do not include spontaneous death in the differential equation for *E. coli* because it is inherently included when measuring the bulk growth rate $\gamma_E$.

Using this experimentally parameterized model, we asked how well the transformation parameters that we measured using purified DNA would predict killing-enhanced HGT in dual-species microbial communities. The model closely matched experimental results, suggesting that DNA released from cells in situ is equivalently available for uptake as DNA purified in vitro (*Figure 4*). Note that counting CFUs in a spot on an agar plate is a destructive measurement, as it requires harvesting and resuspending the entire spot, so successive data points in a time course are actually from different individual spots that were seeded at the same time from the same cell mixture.

## Exploration of environmental impacts on HGT

Experimental measurement of HGT is time-consuming and labor-intensive, limiting the study of how it is affected by environmental conditions. However, our experimentally parameterized, mechanistic model allows us to simulate a much wider variety of conditions. This allows us to predict the most conducive conditions for HGT, the conditions in which contact-dependent killing plays an important role, and what strategies are most likely to inhibit HGT, and thus the spread of MDR.

First, to determine the effect of bacterial seeding density on the fraction of transformed *Acinetobacter*, we simulated surfaces seeded with varying densities of the two species and incubated for two hours. Wild-type *Acinetobacter* had the highest transformation frequency when both species were seeded at higher densities, allowing maximal contact (*Figure 5a*). This can be understood by considering that the killing rate, and thus DNA release, depends on the product of the cell counts of

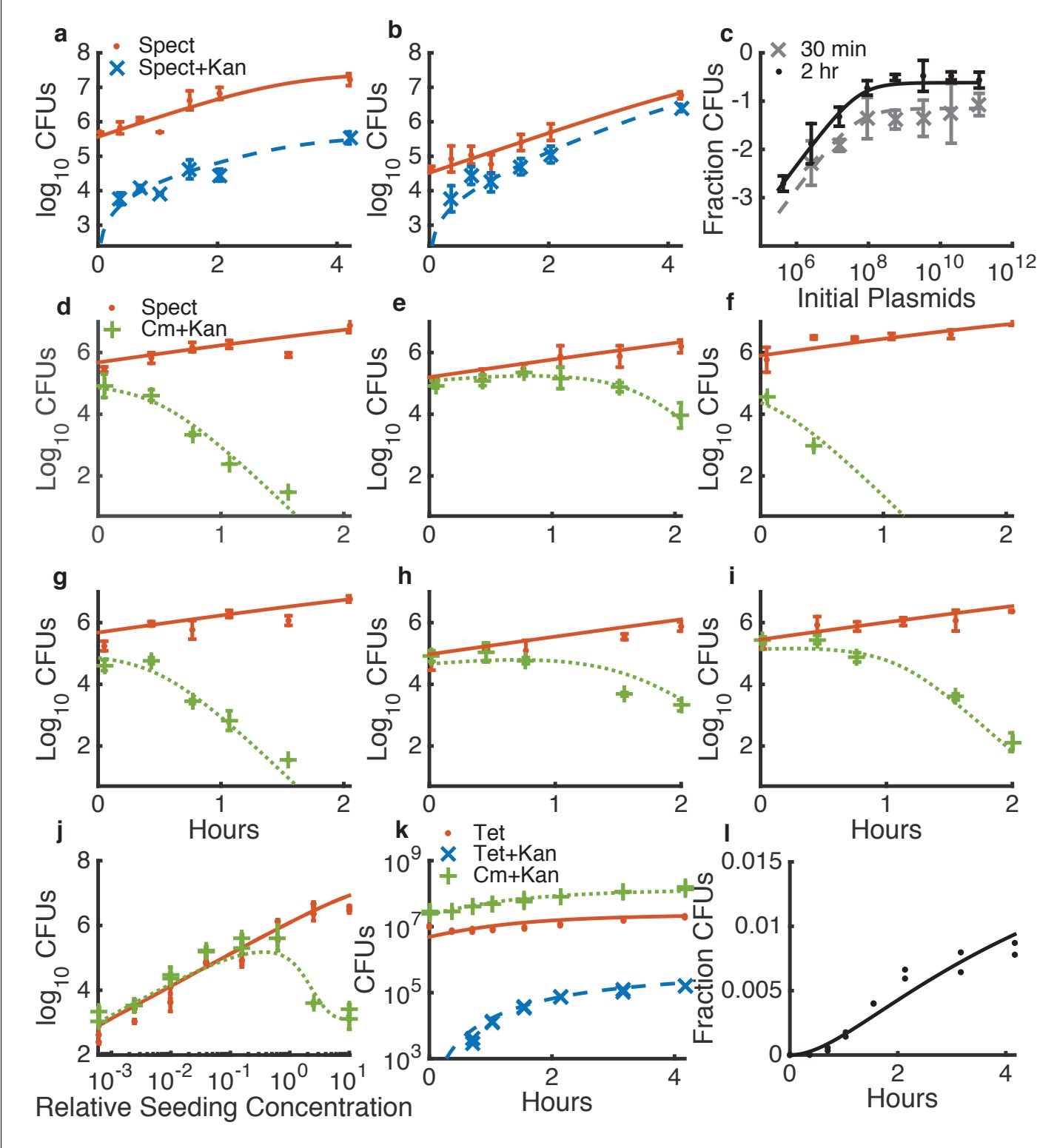

**Figure 3.** Determining quantitative parameters for natural transformation and contact-dependent neighbor killing. Error bars indicate measurement standard deviations of experimental data for a single spot harvested from an agar plate, and lines represent simulations using the shared best fit parameters (*Table 1*). Note all y-axes are log scale except for l. Orange lines indicate *Acinetobacter* (spect-, or in k, tet-resistant), blue lines indicate transformed *Acinetobacter* (additionally kan-resistant), green lines indicate *E. coli* (cm-resistant), and black or grey lines indicate the fraction of *Acinetobacter* that have been transformed. (a–c) Transformation of *Acinetobacter* via homologous recombination of exogenous DNA (pRC03H-2S)

*Figure 3 continued on next page*

*Figure 3 continued*

added just before spotting the cells. (a,b) Time courses of transformation of *Acinetobacter* mixed with limiting (a) or saturating (b) plasmid DNA. (c) Fraction of *Acinetobacter* transformed by a DNA dilution series, harvested after either 30 min or 2 hr. (d–j) Measuring killing of *E. coli* by *Acinetobacter*. (d–i) A selection of independent time courses seeded with varying densities of *E. coli* and *Acinetobacter* (see also *Figure 3—figure supplement 3*). (j) A contact-dependent killing assay using a dilution series of *Acinetobacter* mixed with *E. coli* at the same ratio, but varying total concentration. (k,l) Time series of the number of CFUs (k) and fraction of transformed killing-deficient *Acinetobacter* Δhcp (l) after growth with *E. coli* on agar plates, used to fit DNA 'leakage'.

DOI: https://doi.org/10.7554/eLife.25950.017

The following figure supplements are available for figure 3:

**Figure supplement 1.** Logistic growth fit for *Acinetobacter* spots on agar starting with various initial CFUs.
DOI: https://doi.org/10.7554/eLife.25950.018
**Figure supplement 2.** Logistic growth fit for *E. coli* spots on agar starting with various initial CFUs.
DOI: https://doi.org/10.7554/eLife.25950.019
**Figure supplement 3.** Fitting contact-dependent killing.
DOI: https://doi.org/10.7554/eLife.25950.020
**Figure supplement 4.** Simulations of contact-dependent killing with and without restriction of killing to perimeter cells.
DOI: https://doi.org/10.7554/eLife.25950.021
**Figure supplement 5.** Effective perimeter cells.
DOI: https://doi.org/10.7554/eLife.25950.022

both species (*Box 1*), so the most DNA is released when both species are at high density. In contrast, the transformation frequency of killing-deficient *Acinetobacter* was mainly dependent on only the *E. coli* seeding density (*Figure 5b*), because DNA release from spontaneous lysis depends only on the number of *E. coli*.

From these results, we calculated how the relative importance of killing for HGT depends on seeding density. We defined the degree to which killing of *E. coli* increases HGT to *Acinetobacter* - that is, the enhancement factor - as the ratio of HGT to WT *Acinetobacter* divided by HGT to the killing mutant *Acinetobacter* in the same condition. This killing enhancement was greatest when

**Table 1.** Microbial community growth, killing, and horizontal gene transfer parameters.
See Materials and methods and main text for details.

| Parameter | Description | Value | Source |
|---|---|---|---|
| $\gamma_A$ | *Acinetobacter* growth rate | 1.34 hr$^{-1}$ | *Figure 3—figure supplement 1* |
| $\gamma_E$ | *E. coli* growth rate | 1.51 hr$^{-1}$ | *Figure 3—figure supplement 2* |
| $K_A$ | *Acinetobacter* growth saturation | $2.6 \times 10^7$ | *Figure 3—figure supplement 1* |
| $K_E$ | *E. coli* growth saturation | $1.7 \times 10^8$ | *Figure 3—figure supplement 2* |
| $r_{kill}$ | Killing rate | 15 hr$^{-1}$ | *Figure 3d–j* |
| $K_{kill}$ | Killing saturation | $9.3 \times 10^5$ | *Figure 3d–j* |
| $K_{E\_kill}$ | Killing saturation prey factor | 17 | *Figure 3d–j* |
| $r_{leak}$ | Spontaneous *E. coli* lysis rate | $9.9 \times 10^{-4}$ hr$^{-1}$ | *Figure 3k,l* |
| $c$ | DNA uptake rate | 60 bp/s | (*Mao and Lu, 2016*) |
| $K_{DNA}$ | DNA uptake saturation | $6.7 \times 10^8$ kb | *Figure 3a–c* |
| $\varepsilon$ | Plasmid transformation efficiency | $6.7 \times 10^{-3}$ | *Figure 3a–c* |
| $p$ | Plasmids per *E. coli* cell | 162 | See Materials and methods |
| $G$ | Plasmid equivalents of genomic DNA per cell | 506 | $= 3.6 \times 10^6 / 9093$ |

DOI: https://doi.org/10.7554/eLife.25950.023

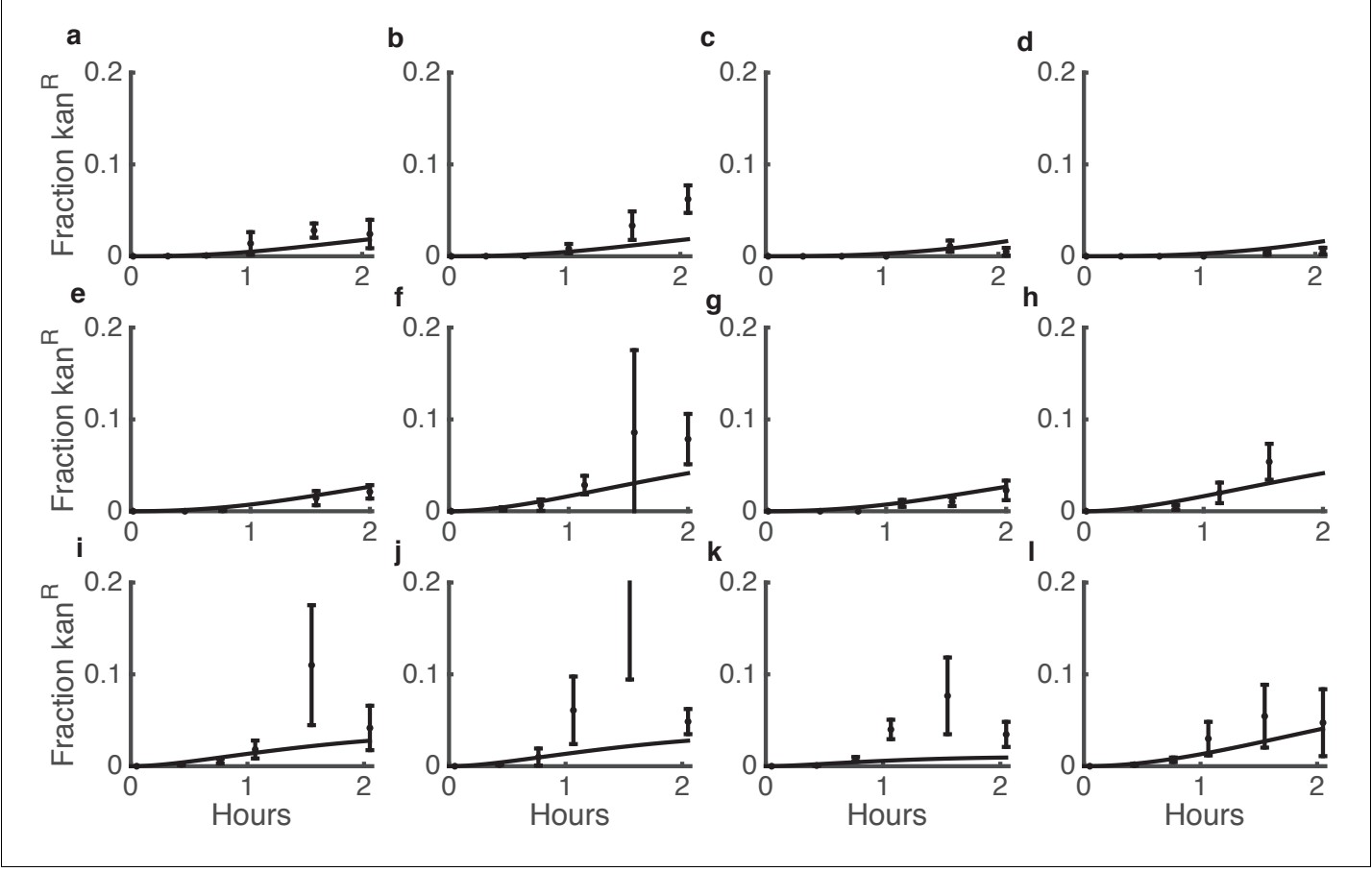

**Figure 4.** Comparison between predicted and actual killing-enhanced HGT in microbial communities. Plotted is the fraction of *Acinetobacter* that have become double antibiotic-resistant due to HGT of kan resistance from *E. coli*. Data are from the same experiments as *Figure 3d–i* and *Figure 3— figure supplement 3*, which shows the total CFUs. Solid lines are model predictions, and error bars are standard deviations of experimental results. Each row of plots is from a different day, and plots within a row are for varying seeding densities of the two species (shown in the Figure Supplement at time 0).

DOI: https://doi.org/10.7554/eLife.25950.024

*Acinetobacter* was seeded at high density and *E. coli* at low density, in which case killing enhanced HGT by nearly 1000-fold (*Figure 5c*).

Next, we used our model to explore how killing-enhanced HGT interacts with incubation time. We simulated a surface seeded with both *Acinetobacter* and *E. coli* at $10^{-3}$ of their respective carrying capacities and calculated HGT to either killing or non-killing *Acinetobacter* over time. HGT increased with time for both the WT and the killing mutant *Acinetobacter* (*Figure 5d*), but the enhancement of HGT provided by killing was greatest within the first few hours (*Figure 5e*). Varying the initial seeding density (*Figure 5f*) revealed that at higher densities, the enhancement factor was greatest immediately after seeding, whereas at lower seeding densities, the enhancement factor did not peak until up to 4 hr after seeding. Killing still enhanced HGT by more than 10-fold for a wide range of seeding densities even after 10 hr (*Figure 5g*), which is long enough to reach the carrying capacity. Overall, these results show that the degree to which contact-dependent killing increases HGT to predatory bacteria is influenced by the total initial cell density, the ratio of the two species, and the length of time that the community has to grow.

In the case of transferring antibiotic resistance genes, it would be desirable to inhibit HGT. Therefore, we used our model to explore how well killing-enhanced HGT could be blocked by two potential environmental perturbations: either degradation by DNase or competitive inhibition with added DNA. First, we determined the DNA degradation rate and amount of competing DNA that would be required to inhibit HGT. We simulated surfaces seeded with both species at $10^{-3}$ of their

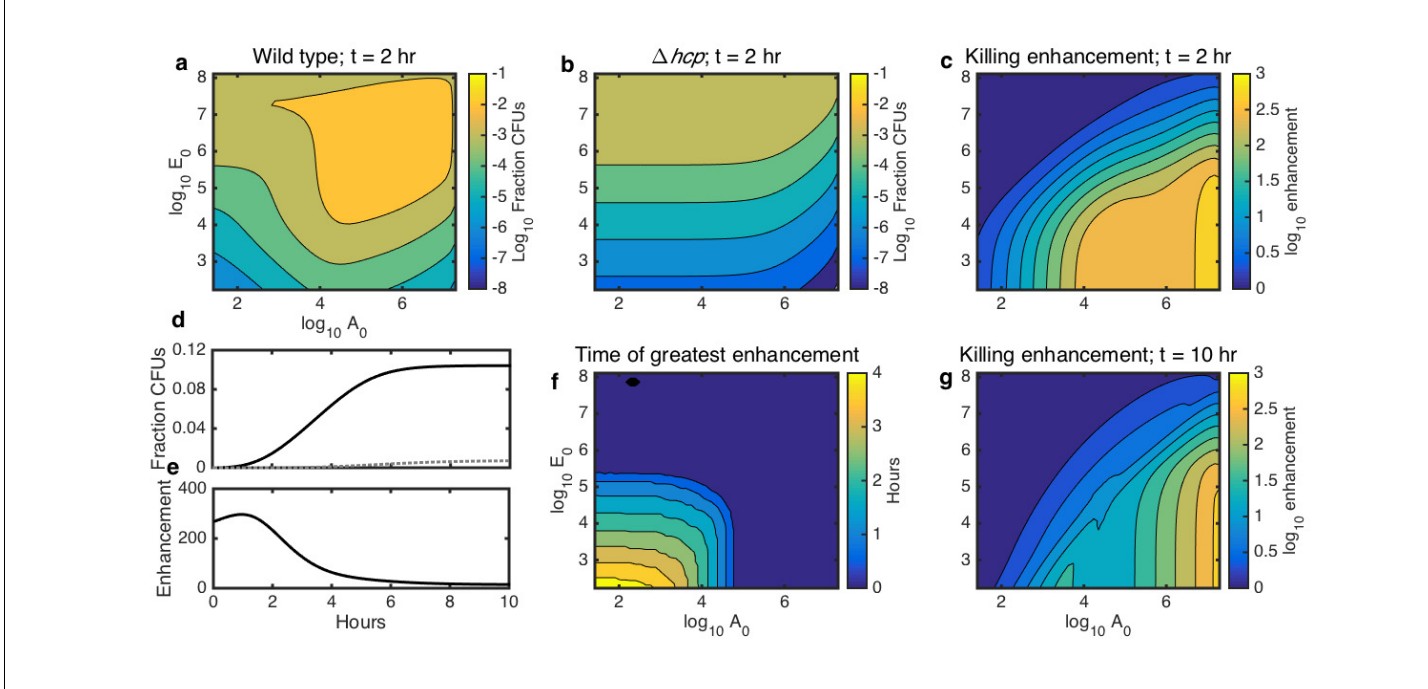

**Figure 5.** Simulations showing the effects of initial species density and interaction time on HGT, and the degree to which contact-dependent killing enhances HGT. The axes in a-c,f,g indicate the initial cell count ($A_0$ and $E_0$) relative to the carrying capacity ($K_A$ and $K_E$) for *Acinetobacter* and *E. coli*, respectively. Contour levels indicate 10-fold changes in a,b, two-fold changes in c,g, and 30 min changes in f. (a) Simulated HGT frequency to wild-type *Acinetobacter* grown with *E. coli* at varying seeding densities for 2 hr. (b) Simulated HGT frequency as in a, but for killing-deficient *Acinetobacter* Δ*hcp*. (c) The HGT enhancement factor provided by killing depending on seeding density; that is, the ratio of HGT to the wild type (a) divided by HGT to the killing mutant (b). (d) HGT frequency over time for agar surfaces seeded with both species at $10^{-3}$ of their respective carrying capacity. Solid line: wild type *Acinetobacter*, dotted gray line: killing mutant Δ*hcp Acinetobacter*. (e) HGT enhancement factor provided by killing over time for the simulation in d. (f) Incubation time at which killing provides the greatest HGT enhancement, for varying seeding densities. (g) HGT enhancement as in c, but after 10 hr, by which time even sparsely seeded communities had approached simulated growth saturation.

DOI: https://doi.org/10.7554/eLife.25950.025

respective carrying capacities and grown for 2 hr. At this seeding density, DNase was effective when it reduced DNA half life to less than about 2–3 min (*Figure 6a*), and competing DNA was effective at about $10^9$ kb or greater (*Figure 6e*).

We then explored how well each condition would inhibit HGT in microbial communities seeded at different cell densities and species ratios. We simulated surfaces seeded with varying numbers of *E. coli* mixed with either killing or non-killing *Acinetobacter*, either with DNA half life fixed to 1 min (*Figure 6b–d*) or with $10^{11}$ kb of competing DNA added at time 0 (*Figure 6f–h*). To quantify the efficacy of each inhibition strategy, we calculated the HGT reduction factor, defined as the ratio of HGT without inhibition to HGT with inhibition. The HGT reduction factor depended on seeding density to varying degrees for both WT and non-killing (Δ*hcp*) *Acinetobacter* for both strategies - degradation with DNase (*Figure 6b,c*) and competitive inhibition (*Figure 6f,g*). Our model had previously predicted contact-dependent killing-enhanced HGT to WT *Acinetobacter* to occur most frequently when both species begin at high density (see *Figure 5a*), so that may be the most important condition in which to inhibit HGT. Importantly, our model predicted DNase to remain effective even at high initial cell density (*Figure 6b*), whereas competing DNA was predicted to dramatically lose efficacy in that condition (*Figure 6f*). This suggests that DNase may better inhibit HGT in dense communities such as biofilms than competitive inhibition with exogenous DNA. Regardless of the inhibition strategy, killing consistently increased HGT to the wild type relative to the killing mutant, even in the presence of the inhibitors (*Figure 6d,h*).

Finally, we experimentally tested the predictions from our model, focusing on cases with relatively high cell numbers where HGT events are detectable. With respect to seeding density, the model predicted experimental results quite well (see *Figure 5a–c*). For WT *Acinetobacter* mixed with *E. coli*

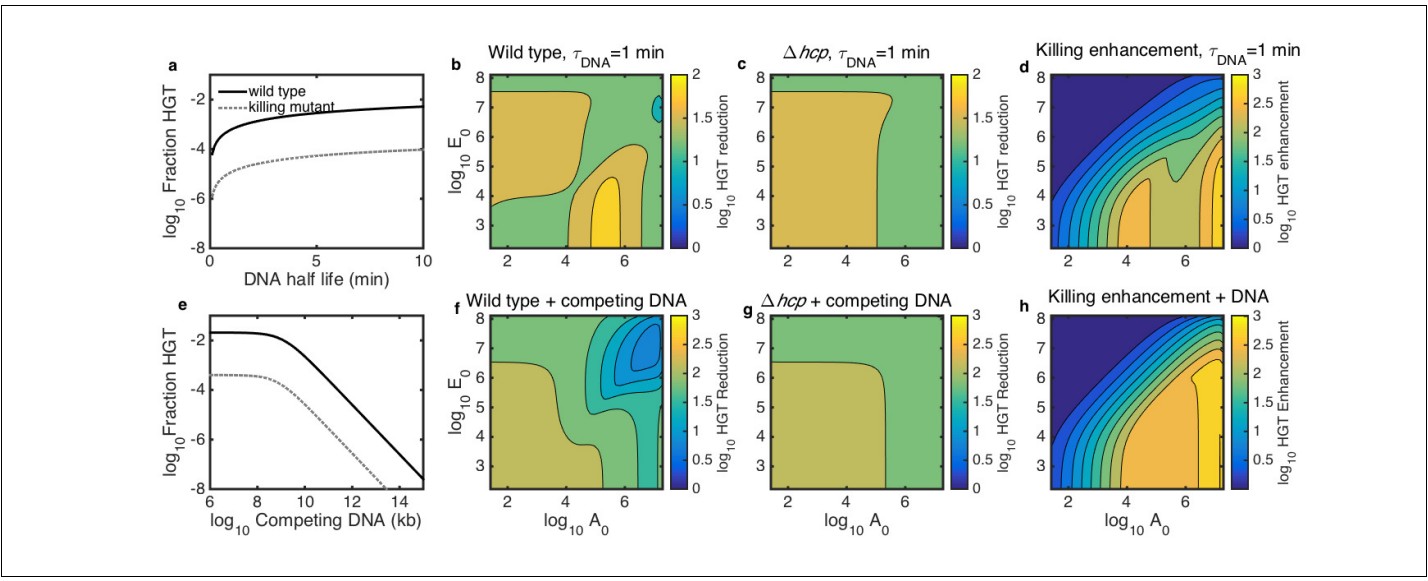

**Figure 6.** Simulated inhibition of HGT by DNase (a–d) or competing DNA (e–h) in simulated microbial communities seeded with both *Acinetobacter* and *E. coli* at $10^{-3}$ of their respective carrying capacities and grown for 2 hr. All contour levels indicate two-fold differences, and all axis and colorbar scales are log10, except DNA half life in a. (a) Fraction of wild type (solid line) and non-killing (dotted grey line) *Acinetobacter* that have undergone HGT as a function of DNA half life in the extracellular space. (b,c) Fraction of wild type (b) and killing mutant (c) *Acinetobacter* that have undergone HGT with the half life of extracellular DNA set to 1 min, for varying initial cell counts. The axes indicate the initial cell count $A_0$ and $E_0$ for *Acinetobacter* and *E. coli*, respectively. (d) The degree to which killing increases HGT as a function of seeding density, with DNA half life set to 1 min; i.e., the ratio of HGT to the wild type (b) divided by HGT to the killing mutant (c). (e) HGT efficiency as in a, but with the addition of varying amounts of competing DNA rather than a finite DNA lifetime. (f,g) Efficiency of HGT as in b,c, but with the addition of $10^{11}$ kb of competing DNA at time 0 rather than DNA decay. (h) Enhancement of HGT provided by killing, as in d, but with $10^{11}$ kb of competing DNA at time 0 and no DNA degradation.
DOI: https://doi.org/10.7554/eLife.25950.026

carrying pRC03H-2S, HGT was greatest when both species were at high initial density, and it decreased along with seeding counts of either species (*Figure 7a*). In contrast, for non-killing *Acinetobacter Δhcp*, the HGT frequency was mainly dependent on the initial number of *E. coli* (*Figure 7b*). The HGT enhancement factor was greatest when the predator was seeded at high density, while the prey was at low density (*Figure 7c*). Also consistent with the model, the HGT enhancement decreased for all seeding densities after overnight growth (compare *Figure 7c* to *Figure 7d*).

Units of DNase are not easily converted to DNA half life on an agar substrate, so we tested a four-fold DNase dilution series (*Figure 7e*) with the two species seeded at approximately equal numbers. As predicted, DNase effectively inhibited HGT even to the wild type at high seeding density (approximately $3 \times 10^6$ CFUs of each species). When cells were seeded at 100-fold lower density, or *Acinetobacter* was killing-deficient, DNase reduced HGT to an even greater extent. All tested levels of DNase reduced HGT below detection (5 CFUs) for *Acinetobacter Δhcp* seeded at the lower density.

In contrast, $10^{11}$ kb of competing DNA was ineffective against HGT to WT *Acinetobacter* at high seeding density (*Figure 7f*). At 100-fold lower seeding density, or when *Acinetobacter* was unable to kill *E. coli*, competing DNA did inhibit HGT. The standard error for these HGT reductions appears relatively large because calculating HGT reduction requires four measurements - total CFUs and transformed CFUs for each of two conditions - and the Poisson-distributed measurement error compounds with each additional measurement (see Materials and methods). Nevertheless, the results were repeatable over 3 separate days, and the figure shows their average.

While our model predicted the key qualitative features of cell seeding density, extracellular DNase, and competing DNA, there were some discrepancies. In particular, HGT inhibition by DNase leveled off above 0.1 units per spot, suggesting a sub-population of *Acinetobacter* with privileged access to released DNA, and inhibition by competing DNA was less than predicted in all conditions. This may be a result of the fact that our model does not capture spatial heterogeneity in DNA concentrations. Lysed *E. coli* release DNA at a locally high concentration, which may provide privileged

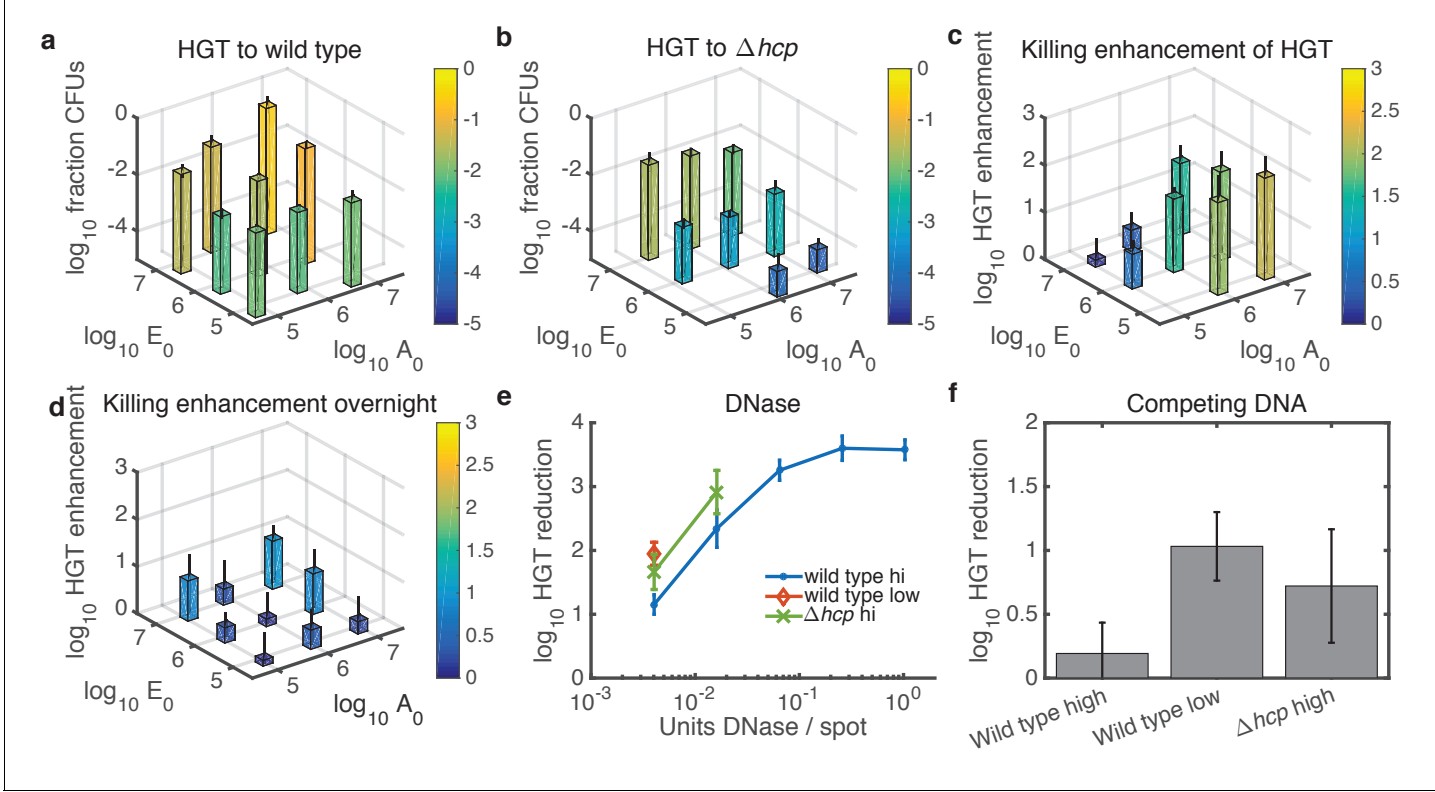

**Figure 7.** Experimental tests of model predictions. (**a–d**) Dependence of HGT on cell seeding density. a HGT frequency to wild-type *Acinetobacter* spotted together with *E. coli* at the indicated cell counts ($A_0$ and $E_0$, respectively) and grown for 2 hr at 37°C. b HGT frequency as in a, but for *Acinetobacter* $\Delta hcp$. The missing bar at the bottom indicates data below detection. (**c**) HGT enhancment provided by killing; i.e., the ratio of data in a to that in (**b**) . (**d**) Same as in c but after overnight growth. (**e**) Reduction of HGT to wild type (blue dot) or $\Delta hcp$ (green X) *Acinetobacter* seeded with *E. coli* at equal high density (approximately $3\times10^6$ CFUs each), or to wild type with both species seeded at low density ($3\times10^4$ CFUs each, orange diamond), all mixed with the indicated amount of DNase before spotting and grown for 2 hr. Missing data points indicate HGT was below detection (5 CFUs), and HGT to *Acinetobacter* $\Delta hcp$ was below detection (5 CFUs) for all tested levels of DNase. Reduction is relative to the same experiment with no DNase added. (**f**) HGT reduction as in e, but adding $10^{11}$ kb of competing DNA before spotting. Error bars indicate the propagated standard error.
DOI: https://doi.org/10.7554/eLife.25950.027

access for directly adjacent *Acinetobacter*. The discrepancies may also reflect the technical difficulty of delivering molecules evenly to real-world communities, which would be exacerbated for mature biofilms.

## Discussion

Humanity's dwindling arsenal of antibiotics is a significant and growing concern. This threat stems from pathogens such as *Acinetobacter* that are able to rapidly accumulate multiple resistance genes, which can make them nearly impossible to treat. High-throughput sequencing has revealed evidence for widespread horizontal gene transfer, but unlike conjugation and phage transduction, we still know relatively little about the microbial dynamics underlying HGT via natural competence (*Mao and Lu, 2016*; *Johnsborg et al., 2007*). It has been observed qualitatively that killing of nearby cells - via fratricide (*Kreth et al., 2005*; *Wei and Håvarstein, 2012*), sobrinocide (*Johnsborg et al., 2008*), or contact-dependent killing (*Borgeaud et al., 2015*) - can enhance HGT, but a lack of quantitative methods has precluded a fuller understanding of the importance of microbial combat in the horizontal spread of genes.

In this paper, we showed that contact-dependent killing by *Acinetobacter* can increase HGT rates from *E. coli* by up to 3 orders of magnitude, making it frequent enough to observe multiple events in real time within microfluidic chips. By subsequently adding kanamycin to our chips, we observed

functionally adaptive emergence of newly drug-resistant bacteria in situ. Significantly, we also showed that killing by one strain in a spatially-structured community makes DNA available for uptake by nearby, non-killing cells. This highlights the role that polymicrobial interactions can play in facilitating HGT. We then developed population dynamic models for both natural transformation and contact-dependent killing, and we fit them to experimental data to obtain biologically relevant parameters. Interestingly, DNA uptake and transformation parameters fit using purified plasmid DNA accurately predicted HGT by DNA released in situ. This was not obvious *a priori*, particularly given that previous experiments have shown extracellular DNA to rapidly lose its transforming ability in real-world conditions (*Mercer et al., 1999*; *Nielsen et al., 2000*; *Nielsen et al., 2007*; *Nordgård et al., 2007*).

For contact-dependent killing, predation can occur only at the perimeter of micro-colonies, where there is contact between predator and prey cells (*Schwarz et al., 2010a*; *Borenstein et al., 2015*; *Hood et al., 2010*; *Schwarz et al., 2010b*; *MacIntyre et al., 2010*; *LeRoux et al., 2012*) (see also Supplemental *Videos 9–12*). In our model, we derived an approximation for the number of cells at the perimeter of their respective colonies, $N^*$ (see *Box 1* and Materials and methods). This allowed us to approximately account for the difference between interior and exterior prey cells, while maintaining the simplicity of a system of ordinary differential equations. This use of $N^*$ to represent perimeter cells yielded a slightly better optimal fit to our data than assuming all cells are equally vulnerable, but the difference was not dramatic. Using all cells $N$ rather than perimeter cells $N^*$, the optimal fit was $r_{kill} = 17$ hr$^{-1}$, $K_{kill} = 1.3 \times 10^6$, $K_{E\_kill} = 21$ (compare to values in *Table 1*), and the best fit sum of squares of residuals increased from 87.0 to 93.9 (calculated using log10-transformed data). See also *Figure 3—figure supplement 4* for a comparison of simulations using the same parameters, fit using the restriction of killing to perimeter cells and shown in *Table 1*, simulated with and without that restriction. The difference between the total number of cells and the number of perimeter cells only becomes significant when the population has grown much greater than the initial cell number (compare solid blue to dashed orange lines in *Figure 3—figure supplement 5*), so we would expect it to be more important in longer time courses. The model presented here can be modified for the case when killing is mediated by diffusible molecules, where there is no distinction between interior and exterior cells in a micro-colony, by simply replacing $N^*$ with $N$.

Using our experimentally parameterized model, we characterized contact-dependent killing-enhanced HGT in a wide range of simulated conditions. Our model revealed that killing is most important for HGT when the prey is at low density, the predator is at high density, and the

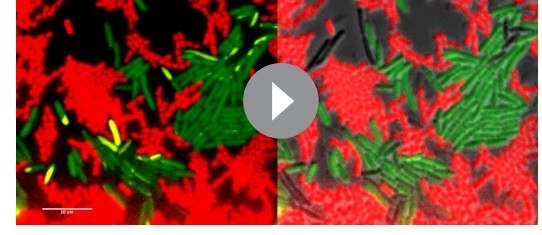

**Video 9.** *E. coli* lysis is concentrated at the periphery of micro-colonies. Four representative micro-colonies were selected from *Videos 2* and *3* to highlight the enrichment of *E. coli* cell lysis at the boundaries of micro-colonies, where the cells are in contact with

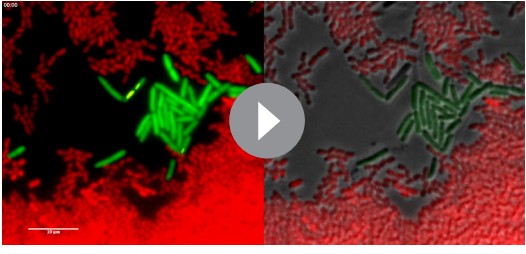

**Video 11.** *E. coli* lysis is concentrated at the periphery of micro-colonies. Four representative micro-colonies were selected to highlight the enrichment of *E. coli* cell lysis at the boundaries of micro-colonies. See caption to *Video 9* for details.
DOI: https://doi.org/10.7554/eLife.25950.030

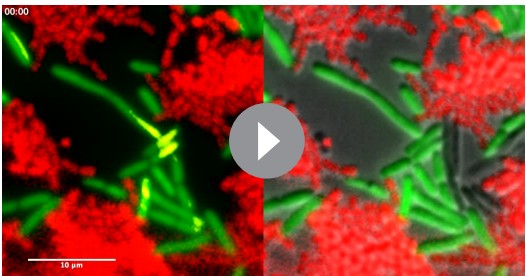

**Video 12.** *E. coli* lysis is concentrated at the periphery of micro-colonies. Four representative micro-colonies were selected to highlight the enrichment of *E. coli* cell lysis at the boundaries of micro-colonies. See caption to *Video 9* for details.
DOI: https://doi.org/10.7554/eLife.25950.031

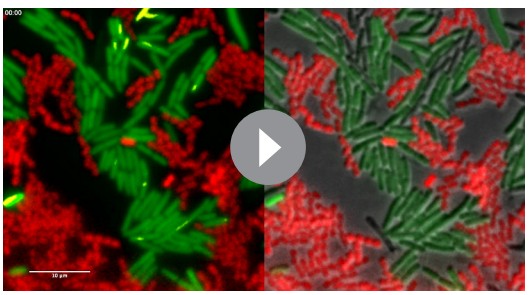

**Video 10.** *E. coli* lysis is concentrated at the periphery of micro-colonies. Four representative micro-colonies were selected to highlight the enrichment of *E. coli* cell lysis at the boundaries of micro-colonies. See caption to *Video 9* for details.
DOI: https://doi.org/10.7554/eLife.25950.029

interaction time is short (*Figure 5c,f*), which was confirmed by experimental data (*Figure 7a–d*). Killing is less important for HGT when the prey outnumbers the predator, because enough prey DNA is released by spontaneous lysis that the additional DNA released by killing no longer provides much advantage. Similarly, killing provides less HGT enhancement when the interaction time is longer, because killing *Acinetobacter* deplete their donor DNA source by killing neighboring prey *E. coli*, whereas the non-killing mutants have a slower, but exponentially growing, source of DNA from spontaneously lysing *E. coli*.

Interestingly, this seeding ratio at which contact-dependent killing most enhances HGT - when the predator outnumbers the prey - is the same condition in which it provides the strongest competitive advantage in terms of cell growth (*Borenstein et al., 2015*). High predator cell density is also the condition that induces fratricide-mediated HGT in *Streptococci* (*Steinmoen et al., 2002*) and simultaneously induces both T6SS expression and competence in *V. cholera* (*Borgeaud et al., 2015*). However, T6SS regulation does not always depend on high cell density, and it can be quite complex and varied, even within strains of the same species, likely reflecting the wide range of ecological contexts and functions performed by the T6SS (*Miyata et al., 2013*; *Bernard et al., 2010*; *Weber et al., 2013*; *Weber et al., 2015*). Given that the only demonstrated advantages are at high cell density, it remains an open question what, if any, selective advantages the T6SS may provide at lower cell density. While T6SS regulation in *A. baylyi* has not been extensively characterized with respect to cell density (*Weber et al., 2016*), it is active and functional in standard in vitro conditions (*Weber et al., 2013*; *Basler et al., 2013*), and we observed T6SS-mediated killing of *E. coli* at all times in the Supplemental *Videos 1–6*.

We also used our model to evaluate the effect of two conditions that may inhibit HGT: addition of either competing DNA or DNAse. Although our model predicted neither condition would eliminated the HGT enhancement provided by killing in our model, both strategies were predicted to reduce HGT when at realistic levels. DNA degradation began to reduce HGT when DNA half life was less than 2–3 min (*Figure 6a*), which is just longer than the half life of free DNA in saliva, about 1 min (*Mercer et al., 1999*). Competing DNA began to reduce HGT at about $10^9$ kb or greater (*Figure 6e*), and given that our agar spots could support $10^7$–$10^8$ cells, that would be 10–100 kb of extracellular DNA per cell at carrying capacity.

Importantly, DNase was predicted to be more effective than competing DNA at high cell seeding densities, which is the condition where contact-dependent killing-enhanced HGT is most efficient. Our experimental results supported this, showing that DNase can reduce HGT to wild type *Acinetobacter* by nearly $10^4$-fold even at high seeding densities, whereas $10^{11}$ kb of competing DNA was ineffective at high cell density. The reduced efficacy of competing DNA in this condition is likely because the high initial killing rate releases a large amount of prey DNA at once, overcoming competitive inhibition.

Interestingly, competing DNA was nearly 10-fold less effective than predicted, which may reflect its physical exclusion from dense microbial communities, or the fact that our model does not reflect the spatial heterogeneity of DNA released from lysing *E. coli*. The potential importance of non-isotropic DNA concentration is also suggested by the observation that there appears to be a small sub-population of *Acinetobacter* with access to prey cell DNA protected from DNase (*Figure 7e*), and the observation that even in three-strain communities, HGT to *Acinetobacter* Δ*hcp* remains slightly lower than HGT to WT cells (*Figure 2c,i*). Spatial dependence of DNA concentration would be a valuable limitation to address in future work; nevertheless, the qualitative trends predicted by our model show its usefulness in guiding experiments and intuition. Additionally, the observation that contact-dependent killing appears to provide a sub-population of cells with privileged access to

DNA that is protected from DNase may help explain how HGT can occur so frequently in real-world environments where DNA is quickly degraded.

While we used *A. baylyi* as a model for the more clinically threatening *A. baumannii*, the two are closely enough related that they are difficult to distinguish with standard phenotypic tests used in the clinic (*Chen et al., 2007*). Indeed, careful re-examination of clinical isolates recently revealed that *A. baylyi* can also opportunistically infect humans, including case clusters in hospitals (*Chen et al., 2008*). It was a clinical *A. baylyi* isolate that contained the first *bla*SIM-1 carbapenemase reported in China (*Zhou et al., 2011*), and the difficulty of distinguishing between the two suggests that *A. baylyi* may in fact be causing additional infections that are either misidentified as *A. baumannii* or identified nonspecifically as *Acinetobacter* (*Chen et al., 2007*, *2008*). Perhaps more importantly, *A. baylyi* may serve as an easily sampled reservoir of genetic diversity for *A. baumannii*, due to the extremely high transformability of *A. baylyi* coupled to the ease with which *Acinetobacter* species exchange DNA. For example, another carbapenemase gene, *bla*OXA23, has been found in multiple *Acinetobacter* species isolated from both humans and animals, including both *A. baumannii* and *A. baylyi* (*Smet et al., 2012*; *Nigro and Hall, 2016*).

In addition to helping to explain, predict, and combat the rapid spread of multi drug-resistance, particularly among *Acinetobacter*, the population dynamics revealed here are likely important in the microbiome and microbial evolution more broadly. Both the T6SS (*Schwarz et al., 2010a*) and natural competence (*Johnsborg et al., 2007*) are widespread among gram-negative bacteria, and both competence and other neighbor killing mechanisms are found in both gram-negatives and gram-positives. Therefore, our results are likely broadly generalizable and may contribute to understanding the known increase of HGT within biofilms (*Madsen et al., 2012*). The method presented here for quantifying natural competence with standardized parameters may also help researchers address other outstanding questions about microbial evolution (*Vos et al., 2015*). In the light of a growing threat from antibiotic resistance, the quantitative methods presented here should greatly aid study of the mechanisms by which bacteria swap genes, shortcut evolution, and outsmart our drugs.

## Materials and methods

### Cell growth, strains, plasmids, CFU counting, and HGT characterization

We obtained *Acinetobacter baylyi* sp. ADP1 from the ATCC and used a lab strain of *E. coli* MG1655. We inserted spectinomycin resistance and mCherry into a putative prophage region of the *Acinetobacter* genome, as described previously (*Murin et al., 2012*). To give *E. coli* genomic chloramphenicol resistance, we first inserted a short 'landing pad' from pTKS/CS, including tetracycline resistance, into a neutral site on the genome (between atpI and gidB), using the recombineering helper plasmid pTKRED (*Kuhlman and Cox, 2010*). We then replaced this landing pad with the chloramphenicol marker from donor plasmid pTKIP-cat, as described previously (*Kuhlman and Cox, 2010*). We cured the cells of pTKRED by growing them overnight with saturating IPTG and arabinose and then screening for spectinomycin sensitivity.

For a replicating bait plasmid, we used the broad-host plasmid pBAV1k (*Bryksin and Matsumura, 2010*), to which we introduced a GFP gene. We derived our genomically integrating plasmids from and the ColE1 origin plasmid pRC03, which replicates in *E. coli* but not in *Acinetobacter*. To add *Acinetobacter* homology to pRC03, we inserted 7 kb of *Acinetobacter* genomic sequence (covering genes ACIAD3424 to ACIAD3429) adjacent to the kanamycin marker, yielding pRC03H. To make a higher efficiency integrating plasmid with genomic homology on both sides of the marker (pRC03H-2S), we moved the kanamycin marker into the middle of the *Acinetobacter* sequence.

To compare acquisition of plasmid vs. genomic resistance genes, we inserted 22 kb of *Acinetobacter* genomic sequence (covering genes ACIAD2681 to ACIAD2697) into the donor plasmid pTKIP-neo and transformed this into the landing pad-containing strain described above. We then used the helper plasmid pTKRED to integrate this sequence in place of the landing pad and finally cured the integrants of pTKRED (*Kuhlman and Cox, 2010*).

We constructed the the non-killing *Acinetobacter* Δhcp by first fusing the tetracycline resistance marker from pTKS/CS to approximately 400 bp homology arms amplified from either side of *hcp* (ACIAD2689) in the *A. baylyi* genome.

For microfluidic experiments, we seeded a custom microfluidic device (*Figure 1—figure supplement 1*) with both *Acinetobacter* and *E. coli* and imaged it on a Nikon TI microscope with an incubated stage set to 37°C and a gravity-driven nutrient flow (LB broth) (*Ferry et al., 2011*). We imaged the experiment shown in *Figure 1a–d* with a 40x objective and that shown in *Figure 1e–k* with a 60x objective. Note that microfluidics with *A. baylyi* are extremely challenging technically, due to its propensity to adhere to and clog the feeding channels (see the horizontal channels at the tops and bottoms of *Videos 1-S6*).

We grew spatially structured communities on LB agar plates (1.5%) in an incubator at 30°C overnight or 37°C for the indicated time. We seeded the communities with 2 ul spots of cell culture, being careful not to introduce bubbles, which can spray aerosolized cells across the plates when they burst. Before seeding, we grew the cells overnight, resuspended them at 1:50 into fresh media, grew them again for 2–3 hr, washed them, and then resuspended again in fresh media. For time-course experiments, the initial cell density can be seen from the CFUs at time 0.

We harvested spots by cutting out spots with a razor blade and resuspending them in 500 ul of PBS buffer. To count CFUs, we made serial ten-fold dilutions of the resuspended cells, spotted 2 ul of each dilution onto selective plates, and counted colonies after incubation overnight. The limit of detetion by this method is $\frac{500}{2} = 250$ CFUs. To lower our limit of detection (*Figure 2b* bars 2 and 4, *Figure 2c* bar 3), we also spread 50 ul of resuspended cells across selective plates, achieving a theoretical limit of detection of 10 CFUs.

To characterize the genomic result of HGT, we picked three clones each of *Acinetobacter* that had acquired kan resistance via either pBAV1k or pRC03H. We isolated any plasmid DNA from these clones and from the bait *E. coli* carrying pBAV1k using a standard miniprep kit (Qiagen, Hilden, Germany), digested with SspI, which should cut twice, and ran the result on a gel (*Figure 2—figure supplement 2a*). To verify genomic integration of the kan gene from pRC03H, we isolated genomic DNA and performed inverse PCR from the kan gene. In particular, we digested the genomic DNA with SalI, diluted to 2 ng/ul, and used T4 DNA ligase to circularize the resulting fragments. Next, we performed PCR using primers pointing outward from the kan gene to amplify the circularized, surrounding genomic region. All three clones had the same PCR pattern (*Figure 2—figure supplement 2b*), and sequencing confirmed that the surrounding region was *Acinetobacter* genome. All enzymes used for cloning were from New England Biolabs (Ipswitch, Massachusetts).

When DNA or DNase was added to the cells, we added it to the cell mixture on ice just before spotting. We used PureLink DNase, resuspended in water at the recommended concentration (2.7 units/ul), frozen in aliquots, and diluted to the desired concentration when needed. For competitive inhibition, we used a high-copy, ampicillin resistance plasmid with a ColE1 origin of replication, which cannot replicate in *Acinetobacter* and does not provide resistance to any of the antibiotics in our experiments (pBest, Promega, Madison, Wisconsin). We grew this in *E. coli*, extracted it with a maxiprep kit, and concentrated it using ethanol precipitation.

## Statistical analysis

Error bars in *Figure 2b–d* indicate the standard deviations of measurements pooled across two culture replicates, each with three measurement replicates. Culture replicates refer to distinct spots seeded with the same cell mixture at the same time. Each resuspended culture was then serially diluted, and these 10-fold dilutions were themselves spotted onto selective plates to count CFUs (measurement spots), as described above. However, the CFUs in each of these measurement spots are expected to be Poisson-distributed, so we spotted each dilution three times, to obtain measurement replicates. Error bars in *Figure 2a* are for three measurement replicates of one culture each. Where data was less than 250 CFUs, only one (larger volume) measurement replicate was counted for each community, as described above. To calculate the pooled variance of HGT frequency across both types of replicates, we followed the following procedure.

1. For each selective antibiotic condition $a$ and each culture replicate $c$, calculate the sample variance of the CFU counts $x$ across measurement replicates, $s_{a,c}^2$.
2. For each culture replicate $c$, calculate the variance of the HGT frequency $f_c = \frac{x_{2,c}}{x_{1,c}}$ using the error propagation formula $s_c^2 = f_c^2 \left( s_{2,c}^2 + s_{1,c}^2 \right)$.

3. Calculate the pooled variance across all culture replicates as $s^2 = \frac{\sum_c \left[(n_c-1)s_c^2 + n_c(f_c-f)^2\right]}{\left(\sum_c n_c\right)-1}$, where the sum is over all culture replicates, $n_c$ is the number of measurement replicates for culture replicate $c$, and $f$ indicates the overall mean of the HGT frequency across culture replicates.

To calculate statistical significance, we performed analysis of variance (ANOVA). In particular, we calculated p-values with the MATLAB function *multcompare*, using the condition means and variances calculated above. Spotting serial dilutions to count CFUs measures data on a log base 10 scale, so to compare data separated by more than one order of magnitude, we first calculated the log base 10 value of each data point (*Figure 2b*, survival of *E. coli* with both strains of *Acinetobacter*, *Figure 2c*, HGT to *Acinetobacter* Δ*hcp* in 2-strain communities). To test for significance where data was below the limit of detection (*Figure 2b*, survival of *E. coli* with WT *Acinetobacter*), we used one-tailed t-tests to determine whether each of the other conditions were significantly greater than that limit of detection (not log-transformed), reporting the largest p-value.

## Modeling HGT in spatially structured communities

In our model (*Box 1*, *Figure 2j*, *Table 1*), both *E. coli* and *Acinetobacter* grow according to the logistic growth equation $\frac{dN}{dt} = \gamma_N N \left(1 - \frac{N}{K_N}\right)$, where $N$ is the number of cells of one species, $\gamma_N$ is the growth rate, and $K_N$ is the carrying capacity for 2 ul spots, and we multiplied the two saturation terms $\left(1 - \frac{N}{K_N}\right)$ to couple the growth saturation of the two species. *Acinetobacter* takes up DNA with maximal rate $c$ and saturation constant $K_{DNA}$, this DNA transforms *Acinetobacter* from species $A_1$ to $A_2$ with efficiency $\epsilon$, *Acinetobacter* kills *E. coli*, thereby releasing DNA, and E. coli leaks DNA at a basal level $r_{leak}$. All variables are absolute numbers, not concentrations. We modeled the DNA uptake and *E. coli* killing terms using Michaelis-Menton kinetics. Since we did all experiments on equal-sized agar spots (resulting from evaporation of 2 ul droplets), we assumed constant volume and combined $V$ with the Michaelis constant $K_{M_{DNA}}$ to obtain a new parameter $K_{DNA}$, with units kb, as demonstrated below for the DNA uptake term:

$$\frac{dD}{dt} = -cA \frac{\frac{D}{V}}{K_{M_{DNA}} + \frac{D}{V}} \tag{1}$$

$$\frac{dD}{dt} = -cA \frac{D}{K_{DNA} + D}; K_{DNA} \equiv K_{M_{DNA}} V \tag{2}$$

Analogously, the killing term $-r_{kill} \frac{A\frac{E}{V}}{K_{M_{kill}} + \frac{A}{V} + K_{E_{kill}}\frac{E}{V}}$ becomes $-r_{kill} \frac{AE}{K_{kill} + A + K_{E_{kill}}E}$, where $K_{kill} \equiv K_{M_{kill}} V$ is unitless.

For contact-dependent killing, predation can occur only at the perimeter of micro-colonies where there is contact between predator and prey cells (*Schwarz et al., 2010a*; *Borenstein et al., 2015*; *Hood et al., 2010*; *Schwarz et al., 2010b*; *MacIntyre et al., 2010*; *LeRoux et al., 2012*) (see also Supplemental *Videos 9–12*), and that perimeter grows proportionally to the square root of the total cell count. Therefore, the relevant quantities in the killing terms are the numbers of perimeter cells, $A^*$ and $E^*$, not the total numbers of cells, $A$ and $E$. Consider a large micro-colony $i$ with $Ni$ cells of radius $r$ and total colony radius $Ri$, where $N_i >> 1$. The number of cells in the micro-colony is the area of the colony divided by the area of a single cell, $N_i = \frac{\pi R_i^2}{\pi r^2}$, so the colony radius is $R_i = r\sqrt{N_i}$. The number of perimeter cells is $N_i^* = \frac{2\pi(R_i-r)}{2r}$, and substituting for $R_i$, we find $N_i^* = \pi(\sqrt{N_i} - 1)$. Each micro-colony grows from a single cell, so assuming equal distribution of cells across micro-colonies and initial number of cells $N_0$, we have $N_i = \frac{N}{N_0}$, so $N_i^* = \pi\left(\sqrt{\frac{N}{N_0}} - 1\right)$. Since there are $N_0$ micro-colonies, the total number of perimeter cells is:

$$N^* = \pi N_0 \left(\sqrt{\frac{N}{N_0}} - 1\right) \tag{3}$$

*Equation 3* is valid for large micro-colonies ($N_i >> 1$), but when colonies consist of only one cell each; that is, $N \leq N_0$, all cells are perimeter cells, so

$$N^* = N \tag{4}$$

Therefore, we define a function for the number of perimeter cells that transitions smoothly between the two limits of *Equation 3* and *Equation 4* at a threshold micro-colony size of 5 (*Figure 3—figure supplement 5*), which we used in the killing term for both $A^*$ and $E^*$:

$$N^* = N\left(\frac{1}{1+\left(\frac{N}{5N_0}\right)^2}\right) + \pi N_0\left(\sqrt{\frac{N}{N_0}}-1\right)\left(\frac{\left(\frac{N}{5N_0}\right)^2}{1+\left(\frac{N}{5N_0}\right)^2}\right) \tag{5}$$

*A. baylyi* natural competence is largely proportional to its growth rate (*Palmen et al., 1993*), so we assumed the DNA uptake slows with the same saturation factor as cell growth. Over a longer time scale, communities on agar plates will continue to grow slowly as fresh nutrients diffuse in and to replace dying cells, but this model focuses on the early dynamics of a growing community in the very early stages of biofilm development.

## Parameter fitting

Transformation of *Acinetobacter* by homologous recombination (HR) is more efficient than via replicating plasmids (*Palmen et al., 1993*), so it allows measurement of HGT with higher sensitivity. Therefore, we measured HGT parameters using pRC03H-2S, which cannot replicate in *Acinetobacter* but contains homology to the *Acinetobacter* genome, allowing *Acinetobacter* to integrate the kan gene via HR.

All parameter fitting (*Table 1*) was performed using the least squares nonlinear optimization function *lsqnonlin* in MATLAB. We fit each component of the model sequentially using simplified experiments. To determine the growth parameters, we measured growth curves for monoculture cell spots on agar and fit them using the logistic growth equation $N(t) = \frac{K_N N_0 e^{\gamma_N t}}{K_N + N_0(e^{\gamma_N t}-1)}$, where $N$ is the number of cells, $\gamma_N$ is the growth rate, and $K_N$ is the saturation level for 2 ul spots (*Figure 3—figure supplements 1* and *2*). Using these growth parameters, we then fit the killing rate by measuring CFUs in co-cultured communities over time and using that data to fit the model as integrated using *ode23* in MATLAB (*Figure 3d–j*). For simultaneous fitting of data from multiple experiments (*Figure 3*, *Figure 3—figure supplement 3*), we defined custom objective functions to calculate the prediction error for all data points, and then minimized the total sum of squares with *lsqnonlin*.

For DNA uptake rate $c$, we used the previously measured 60 bp/s (*Palmen et al., 1993*). To measure the other key HGT parameters $K_{DNA}$ and transformation efficiency $\epsilon$, we seeded monoculture *Acinetobacter* spots on agar with known numbers of plasmids, and measured HGT over time. We simultaneously fit the HGT curves for time course and serial dilution data to the model as integrated using *ode23* (*Figure 3a–c*). We measured plasmids per *E. coli* cell by growing a shaking tube of cells to exponential phase, counting cell density by spotting serial dilutions, isolating plasmid DNA with a standard miniprep kit, measuring obtained DNA density with a spectrophotometer, and converting ng of DNA to plasmids per cell. While plasmid copy number may vary as conditions change (e.g. growth phase and liquid vs. solid media), most of our results present log-scale effects, so this should not affect the qualitative conclusions. For genomic DNA per cell $G$, we divided the *E. coli* genome (3.6 Mb) by the plasmid length (9093 bp) to obtain the number of plasmid equivalents of genomic DNA.

## Simulations

Full simulations of contact-dependent killing and HGT in microbial communities were performed by integrating the population dynamic model with the MATLAB function *ode23*, using parameters measured as described above for pRC03H-2S. For DNA, we defined $D$ to be the number of plasmids, so we converted the units of $c$ and $KD$ using the conversion factor 9093 bp/plasmid.

## Supplemental note 1

In *Videos 1–6* and *9–12*, GFP expression in *E. coli* is poorly repressed early in the movies, and mCherry expression in *Acinetobacter* becomes bleached later in the movies. We believe both of these phenomena are likely due to nutrient restriction, caused by *Acinetobacter* adhesion, growth,

and eventual clogging in the channels meant to supply nutrients (see the top and bottom strips in the Supplemental Movies). Indeed, in our experience, the stickiness of *Acinetobacter* makes microfluidic experiments much more challenging than with only lab *E. coli* strains.

LacI repression in *E. coli* is less complete when nutrients are limiting (*Grossman et al., 1998*). The *E. coli* in *Videos 1–6* are likely nutrient-deprived because *Acinetobacter* are coating the channels, both consuming nutrients and reducing flow rate. In *Videos 7–8*, on the other hand, the media had been switched to include kanamycin, which inhibited all *Acinetobacter* that had not acquired resistance via HGT from *E. coli*. This is likely why the GFP repression in *E. coli* is less complete in *Videos 1–6* than in *Videos 7–8*.

Conversely, *Acinetobacter* appear to reduce expression of mCherry as they become more nutrient-limited. We speculate that mCherry fluorescence fades while GFP remains visible in *Acinetobacter* because (i) the mCherry gene is single-copy on the genome while the GFP plasmid is multi-copy, (ii) mCherry bleaches more readily and has lower intrinsic brightness than GFP (*Shaner et al., 2005*), and (iii) there may be promoter, ribosome-binding site, and codon usage effects affecting expression in *A. baylyi*, which is not nearly as well-characterized as *E. coli* in these respects.

Regardless, in the Supplemental *Videos 1–6*, it is clear when and where all *E. coli* are killed, because the GFP disappears suddenly within a single time-frame, indicating lysis as opposed to gradual bleaching. For GFP to re-emerge in an area where all *E. coli* have been killed (white circles in *Videos 1–6*), it must be expressed by the remaining *Acinetobacter*.

## Acknowledgements

RMC was supported by a fellowship from the Hartwell Foundation. The authors would like to thank Bart Borek for valuable discussion regarding modeling, Ameen Rahimi for assistance in constructing the integrating plasmid pRC03H-2S, and Life Science Editors for assistance in editing the manuscript.

## Additional information

### Funding

| Funder | Grant reference number | Author |
| --- | --- | --- |
| Hartwell Foundation | | Robert M Cooper |
| National Institute of General Medical Sciences | San Diego Center for Systems Biology (P50 GM085764) | Robert M Cooper Lev Tsimring Jeff Hasty |

The funders had no role in study design, data collection and interpretation, or the decision to submit the work for publication.

### Author contributions

Robert M Cooper, Conceptualization, Data curation, Software, Formal analysis, Funding acquisition, Validation, Investigation, Visualization, Methodology, Writing—original draft, Writing—review and editing; Lev Tsimring, Resources, Formal analysis, Supervision, Funding acquisition, Project administration, Writing—review and editing; Jeff Hasty, Resources, Formal analysis, Supervision, Funding acquisition, Project administration

### Author ORCIDs

Robert M Cooper  https://orcid.org/0000-0003-2136-0403

### Decision letter and Author response

Decision letter https://doi.org/10.7554/eLife.25950.034
Author response https://doi.org/10.7554/eLife.25950.035

## Additional files

### Supplementary files

• Source code 1. Supplementary Matlab model file used to generate *Figures 6* and *7*. This uses parallel processing - if you would rather not, please replace *parfor* with *for*. It takes about 15 min to get all the way through on my desktop computer using parallel processing.
DOI: https://doi.org/10.7554/eLife.25950.032

• Transparent reporting form
DOI: https://doi.org/10.7554/eLife.25950.033

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
