## [Decision Letter]

Thank you for submitting your article "Microbiome population dynamics enhance horizontal gene transfer and spread of multi drug-resistance" for consideration by *eLife*. Your article has been reviewed by three peer reviewers, and the evaluation has been overseen by a Reviewing Editor and Gisela Storz as the Senior Editor.

The reviewers have discussed the reviews with one another and the Reviewing Editor has drafted the following issues that we feel you must address if this work is to be considered for publication in *eLife*. At this point, please respond with an experimental plan and a time table for the completion of the work. The Board and reviewers will then consider your plan and issue a binding recommendation.

Summary:

The authors report horizontal gene transfer (HGT) to *Acinetobacter baylyi* and show that bacterial contact-dependent predation by *A. baylyi* via the type VI secretion system can enhance cross-species horizontal acquisition of antibiotic resistance genes by three to four orders of magnitude. Real-time observations show predator *A. baylyi* cells functionally gaining multi-drug resistance through functional, adaptive, horizontal gene transfer through natural competence by lysing neighboring prey cells and horizontally acquiring the released genes. The authors then present a dynamic mechanistic population-based model to quantify this killing-enhanced gene-transfer, predict its prevalence in different conditions, and evaluate strategies to reduce it.

Essential revisions:

1) All reviewers were concerned that the authors were overselling the clinical relevance of the work (given that *A. baylyi* is not a clinically important organism) and agreed that comments about the microbiome +biofilm aspects of the work were also overblown. At the very least these need to be toned down and some important caveats added, as the work concerns a microbiome only in a very minimal sense.

2) A major strength of the work was the combination of the experimental work and the model, but the reviewers felt the link between the two was weak and should be strengthened, ideally by testing model predictions. For example, there are predictions about the role of the predator/prey ratio and the effects of DNAse or competing DNA. Experimental validation of at least one aspect of model predictions would considerably strengthen this work (of course, it might also highlight a problem with the model, another good reason for doing it).

3) The methods were lacking precision, with insufficient detail provided to allow the work to be replicated (see https://elifesciences.org/content/5/e21070 and comments from reviewer #3).

4) It is not completely clear what is going in all the videos and stills. This needs to be clarified (see comments from reviewer #2).

5) There are well-reasoned objections to the term "proximicide". A better-fitting term should be substituted.

6) The assumption that "predation can occur only at the perimeter of micro-colonies" needs to be supported by some images/data. It would also be helpful to compare model fits to data with and without this assumption.

7) Section 4.1 ninth paragraph reports adding DNAse to block HGT, but experimental data needs to be supplied (reviewers failed to find them anywhere).

8) The manuscript should extend the description of previous work on fratricide in Streptococcus and competence-mediated neighbor predation and transformation (see comments from reviewer #3).

The broader importance of the work should also be discussed (see comments from reviewer #4).

9) The reviewers had two concerns about the title: first, this paper is about the microbiome only in a very minimal sense. Second, the focus in the title on multi-drug resistance suggests a clinical relevance that reviewers felt was lacking, given the focus *A. baylyi* (a soil organism) and no good reasons for thinking the results would be directly relevant for *A. baumanii*. A revised title was felt to be appropriate.

*Reviewer #2:*

1) Section 4.1 first paragraph. I don't understand why that first result is 'unexpected'. Is it because the plasmid doesn't have an origin or replication so should not be stably maintained in *Acinetobacter*, or because the time would be too short to expect transfer? Could they explain this.

2) In Video 1–Video 6 it is really hard to tell which bacteria is *E. coli* and which is *Acinetobacter*. The background fluorescence of the green cells at the start of the video (presumably these are *E. coli*, since they subsequently lyse and *Acinetobacter* should not be green yet) is about as high as the 'positive' green at the end of the video in *Acinetobacter*, so the lacI repression doesn't seem very strong. The authors say that at the end the bacteria are red and green, but in the videos I couldn't see any red + green bacteria – the red all looks bleached so the bacteria are only green. How do we know that these are *E. coli*? Would it be possible to trace the lineage back through the movie (only for a few, as an example) to show that these green *Acinetobacter* micro colonies really started as red cells, and better still that they were physically located close to an *E. coli* cell that subsequently lysed?

3) Section 4.1 ninth paragraph, they mention adding DNAse to block HGT – where is this data? I couldn't find it. This is a good control.

4) From the modelling part of the paper they come up with some testable predictions about the role of ratio of predator/prey and the effects of DNAse or competing DNA. Given that they have a really straightforward quantitative assay in their hands, I am wondering why they did not experimentally test some of these hypotheses. I understand the microfluidics is technically challenging, but presumably it would be straightforward enough to repeat their drug resistance acquisition experiments from Figure 2 using a few of the predicted variables?

5) Intuition behind many of the modelling results is discussed, but I think it could be worth doing this a bit more thoroughly in the appendix. Often the most useful thing about models is helping to develop this intuition.

*Reviewer #3:*

In the manuscript by Cooper et al. the authors describe horizontal gene transfer in *Acinetobacter baylyi*. Precisely, the author investigated the propensity of this organism to use kin-discriminated neighbor predation coupled to competence-mediated DNA uptake to enhance HGT. Such enhancement of HGT by killing of adjacent cells followed by the absorption of their DNA has previously been described for Gram-positive bacterial strains of the genus Streptococcus in which case competence cells attacked their not yet competent siblings (known as fratricide; note that kin-discriminated killing is also possible in this context). The Gram-negative pathogen Vibrio cholerae, on the other hand, was shown to use a molecular killing device (the type VI secretion system; T6SS) to attack its non-kin neighbors, which is followed by the uptake of the prey-released DNA. Experimental data showed a significant enhancement of HGT through the regulatory link between kin-discriminated killing and DNA absorption and this earlier study concluded that the T6SS enhances HGT in this organism. The current study follows up on these initial observations as it describes a similar link between neighbor predation and transformation for *A. baylyi*. The authors move on to describe a quantitative model, which takes predator and prey growth rates, natural transformability, cell density and DNA stability into consideration. Through the use of this model, the authors concluded that competing DNA would be less efficient to interrupt HGT under such killing-enhanced HGT conditions, while DNases might be useable for intervention strategies.

While the model is very interesting and the study nicely contributes to our basic knowledge on HGT, the manuscript tries to sell this in the context of the microbiome (microbiota?), biofilm, and the spread of antibiotics resistance in hospital-acquired pathogens (see title). While all of these areas are extremely important, this study does not address any of these areas. Indeed, *A. baylyi* is a soil organism and neither a part of the microbiota nor a problem in hospitals (while *A. baumannii* is, but this organism was not studied here). Thus, sentences like "Therefore, expression of GFP in Acinetobacter should indicate a newly kanamycin-resistant strain, whose de novo appearance demonstrates the clinical relevance of HGT within a microbiome community" seem totally out of place. In general, using key words such a multi-drug resistance (MDR) strains over and over throughout the manuscript seems highly inappropriate. Indeed, all that was shown here are in vitro selected strains in which a single antibiotics gene was transferred by the uptake of a plasmid. This method is commonly used in the transformation field and nobody ever considers such strains as "MDR". The provided experimental data would likewise work if the transfer of a metabolic gene was scored in auxotrophic mutants instead of the transfer of a resistance genes (which is, in fact, only a tool to enumerate transfer efficiencies of any single gene).

Another problem is that the title of the manuscript implies information on population dynamics of the microbiome (=microbiota is probably meant). However, the work describes the transfer of plasmids between a laboratory *E. coli* strain and the soli bacterium *A. baylyi*. Thus, the stretch to the microbiota is far-fetched. Lastly, the work also does not address bona fide biofilms, which is also highly misleading.

In summary, the model part of the study is very interesting and provides novel information on the dynamics of HGT linked to neighbor predation while the experimental part is of limited novelty and lacks methodological information (an extensive list is provided below).

1) *Acinetobacter baumannii* is indeed an opportunistic pathogen and part of the "ESKAPE" pathogens and therefore listed under Centers for Disease Control and Prevention, 2013, (and in a recent report by the WHO) as major threat because of its high level of antibiotic resistance. However, *Acinetobacter baylyi*, which was used in the experimental settings of the current study, is primarily a soil bacterium with very few studies that indicate that it could also be a causative agent of hospital-acquired infections. Thus, stressing that antibiotic resistance is such a problem in hospital settings and that multi-drug resistance rates exceed 60% in *Acinetobacter* is too generalized and misleading. Indeed, this is primarily the case for *A. baumannii*, which is also solely referred to in Antunes, Visca and Towner, 2014. Consequently, the authors should explain to the reader why the experiments were done in *A. baylyi* instead and they should provide valid arguments why this might anyway be applicable to hospital-acquired pathogens and especially *A. baumannii*. Using a different non-pathogenic species of the same genus and then concluding that this is also relevant for the pathogenic strains seems inappropriate (e.g., nobody would solely rely on data on the non-pathogenic bacterium *Mycobacterium smegmatis* to conclude anything of significant importance about the pathogen *Mycobacterium tuberculosis*). Thus, the authors should provide comparable data for *A. baumannii* or clearly state that *A. baylyi* was used as a surrogate for pathogenic *Acinetobacter*. In the latter case, the manuscript needs to be significantly toned down – especially the Abstract – with respect to the role in antibiotic transmissions in hospitals etc. Indeed, at the start of the last paragraph of the Introduction, "In this work, we extend the observation of killing-enhanced HGT to the clinically threatening *Acinetobacter* genus" is inappropriate, as most species within the genus *Acinetobacter* are not at all clinically threatening but soil inhabitants.

2) Title "Microbiome Population Dynamics" and Results: "enabling visualization of model microbiomes" – this is irrelevant here, as *E. coli* alone is not representative of "microbiomes" (even though the *E. coli* strain was probably a descendant of the original *E. coli* K12 strain, which was indeed isolated from a stool sample of a patient in 1922) and as far as I am aware of *A. baylyi* has never described as members of the microbiome because it is in fact a soil organism as stated above (if it is a common member of the microbiota, a reference is for sure needed). Thus, this must be considered as "overselling" and the link with the microbiome is non-existent (except if every *E. coli* study would be likewise considered a microbiome study, in which case old conjugation papers on F factor would have already shown horizontal gene transfer in the microbiome).

3) If the study is meant to be meaningful for *A. baumannii* and its high prevalence of antibiotic resistance genes, the manuscript should contain the information in T6SS regulation in this organism (e.g., Weber et al., 2015, PNAS). This study shows that multidrug-resistant strains of *A. baumannii* harbor a self-transmissible resistance plasmid that negatively regulates the T6SS. Thus, following HGT of this plasmid the T6SS is inactive and could have significant consequences, which should be discussed. Along the same line, the authors should provide information about T6SS regulation in *A. baylyi*.

4) The manuscript should extend the description on previous work on fratricide in Streptococcus and competence-mediated neighbor predation and transformation in *Vibrio* (=same T6SS-mediated killing procedure, as addressed in the current study). In fact, the finding in Figure 2 that T6SS enhances HGT and that "elimination of proximicide impaired HGT by about 100-fold" was shown before for *V. cholerae* for T6SS-mediated neighbor predation and transformation. Notably, several other experiments such as the nuclease control experiment, the testing of T6SS-mutants, the transformation with purified DNA, the homologous-recombination of genomic DNA that contains homolog regions etc. are likewise the same as in the previous study. Thus, the link between neighbor predation and DNA transfer is not new. The quantitative model, on the other hand, is extremely interesting and should represent the central part of the current manuscript. It is also recommended that the authors discuss whether their model would also be applicable to the previously studied kin-discriminating neighbor-predating bacteria (that is, *Streptococci* and *Vibrios*).

*Reviewer #4:*

This is an intriguing and broadly interesting paper, supported well by multimedia (video) presentations. Microbial ecology was long a relatively niche field, but with paper after paper finding a general audience, it is in ascendance. While key findings (horizontal gene transfer [HGT] via/following contact antagonism [CDI]; HGT linked to antagonism) were developed initially in the *Vibrio* and *Streptococcus* systems, the new finding provides the required generalization to interpret the phenomena as broadly applying to Gram negative bacteria, including a wide swath of environmental and likely pathogenic bacteria; and the new data is mechanistically detailed.

The new finding addresses longstanding problems in microbial evolution about the amazing rate (and heterogeneity) of HGT. It does not necessarily address the question of why a cell would want to take up the DNA of a newly dead cell, which was clearly not well served by it; but there are hypothetical answers to that. For example, if relatively untargeted proximity killing (say, by physical means) is orthogonal to the traits which increased the frequency of the cell in community or population, then successful cells will be killed more frequently and their DNA may prove valuable. This is particularly interesting under bacteriostatic conditions, such as nutrient limitation and antibiotic exposure.

The experiments are carefully done. While I typically dislike 'and then we' papers and presentations, there is an element of surprise in the initial finding that justifies some of that narrative structure. Perhaps further findings would be better organized outside of chronology; but this is stylistic. The demonstrations are powerful and effective.

The mathematical models of HGT via CDI are not necessarily as compelling as the experimental data because of the reality gap between even in vivo and in the environment(s). Even where this was partly closed, the math was left somewhat hanging. As an example, the presence of extracellular DNAse effectively reduces the rate of HGT. This was tested in experiment, and performed as a mathematic model, but I don't see a clear connection made in the text between the model and data demonstrating the effect. I similarly didn't see a clear connection between the competing DNA model and an experiment demonstrating the fit or lack of fit to the model. For the conclusions about likely therapeutic strategies, at least some in vitro data should be clearly fit to the model.

Natural interest in MDR aside, this research has broader importance and the discussion should reflect that. How does this tie into the local taxonomic and genomic heterogeneity of natural populations? Soil, healthy microbiome, unhealthy microbiome? What sort of genes are likely to be exchanged (siderophore uptake, AT pairs, adhesins, phage resistance)? Clearly DNA transfer isn't restricted to a single species; yet homology was demonstrated as accelerating the integration of material into the genome. Some commentary on this is appropriate.

In conclusion, the writing is generally good and the methods, particularly the microfluidics, are clever and elegant. The data presentation avoids many potential pitfalls and supports most of the Discussion. There are a couple points that could be strengthened, particularly the connection between the DNAse experiments and the model; and the Discussion should take a more global microbiology perspective.

---

## [Author Response]

Essential revisions:1) All reviewers were concerned that the authors were overselling the clinical relevance of the work (given that A. baylyi is not a clinically important organism) and agreed that comments about the microbiome +biofilm aspects of the work were also overblown. At the very least these need to be toned down and some important caveats added, as the work concerns a microbiome only in a very minimal sense.

Thank you for raising these points – it is clear that we need to better justify the model system studied here and be more careful with terminology. There appear to be three main issues here:

i) Extrapolation from *A. baylyi* as a model to implications for *A. baumannii*, ii) Use of the word “microbiome”, iii) Use of the word “biofilm”.

The common thread among these concerns seems to stem from our extrapolation from simplified, bottom-up, model systems to more complex real-world scenarios. We agree that there are important distinctions that we need to make more clear, but we also believe that bottom-up mechanistic studies like this one are a necessary complement to top-down studies of more complete systems. In the specific case of HGT, top-down microbiome sequencing has revealed that HGT is remarkably rapid in *Acinetobacter*, but the exact mechanisms and specific inter-species dynamics within biofilms that lead to this high-level observation can only be elucidated with bottom-up studies of simplified model systems. Because quantitative, mechanistic understanding of “biofilm” and “microbiome” dynamics requires isolating particular interactions in minimal systems, we do not believe it is inappropriate to discuss extrapolation to the more complete systems, although we agree we should make this link and caveats more clear.

i) Extrapolation from *A. baylyi* as a model to implications for *A. baumannii*. It is clear we should place stronger emphasis on the fact that *A. baylyi* is not as immediately clinically relevant as *A. baumannii*, and indeed it is primarily known as a soil organism. However, we expect that the same neighbor killing-enhanced HGT dynamics observed in *A. baylyi* should be generalizable to *A. baumannii*, particularly given their close relationship and the fact that they (along with many other gram-negatives) share the two key features of T6SS-mediated killing and natural competence. We have added discussion both in the Introduction:

“As a model system, we used *A. baylyi*, which is closely related to *A. baumannii* [Peleg et al., 2012], because it is genetically tractable, fully sequenced, and well-studied [Elliott and Neidle, 2011]. The two also share the key features of T6SS-mediated killing [Weber et al., 2013; Carruthers et al., 2013] and natural competence [Soledad Ramirez et al., 2010; Wilharm et al., 2013], and they are similar enough that standard phenotypic assays used in the clinic often fail to distinguish between them [Chen et al., 2007].”

And a more detailed discussion in the Discussion:

“While we used *A. baylyi* as a model for the more clinically threatening *A. baumannii*, the two are closely enough related that they are difficult to distinguish with standard phenotypic tests used in the clinic [Chen et al., 2007]. […] For example, another carbapenemase gene, blaOXA23, has been found in multiple *Acinetobacter* species isolated from both humans and animals, including both *A. baumannii* and *A. baylyi* [Smet et al., 2012; Nigro and Hall, 2016].

ii) Use of the word “microbiome” Thank you for pointing out this error – “microbiota” or “microbial” would be the more appropriate term in most of the places where we used it. We have changed the title and the word “microbiome” now appears only twice, in discussions of broader implications in the Abstract and the final paragraph of the Discussion.

iii) Use of the word “biofilm” It is true that our work focused on only the very early stages of biofilm formation, but we use the term “biofilm” to indicate that cells are surface-attached and developing a spatially-structured community, rather than in shaking culture tubes. We also note that the term “biofilm” has previously been used to describe simplified microbial communities with spatial structure, for example, 2-dimensional, single-species cultures in a microfluidic chip (Liu et al. 2015 Nature). We have clarified this more explicitly in the Results:

“While these experiments captured only the beginning stages of biofilm development, we use the term biofilm to indicate that the cells were surface-attached and developing a spatially-structured community, rather than growing in shaking culture tubes.”

2) A major strength of the work was the combination of the experimental work and the model, but the reviewers felt the link between the two was weak and should be strengthened, ideally by testing model predictions. For example, there are predictions about the role of the predator/prey ratio and the effects of DNAse or competing DNA. Experimental validation of at least one aspect of model predictions would considerably strengthen this work (of course, it might also highlight a problem with the model, another good reason for doing it).

This is an excellent suggestion. We have performed these experiments and the results are shown in Figure 6 and discussed in the subsection “4.3 Exploration of Environmental Impacts on Horizontal Gene Transfer” and the Discussion.

3) The methods were lacking precision, with insufficient detail provided to allow the work to be replicated (see https://elifesciences.org/content/5/e21070 and comments from reviewer #3).

Thank you for pointing out the areas that were not clear; we have added more discussion to the manuscript as requested.

4) It is not completely clear what is going in all the videos and stills. This needs to be clarified (see comments from reviewer #2).

Thank you for pointing out these areas. We have clarified these points as requested.

5) There are well-reasoned objections to the term "proximicide". A better-fitting term should be substituted.

We recognize the potential for confusion, as sister cells are immune to killing, thus not all cells in the immediate “proximity” are killed. However, appropriately-defined short terms are often valuable aids for discussion of complex phenomena. As a substitute for contact-dependent lysis of unrelated directly adjacent cells, we feel “proximicide” improves readability. We have added the following text to be more clear about how we are using the proposed term, but if the reviewers and editors still feel it is not useful, we could remove it.

“Given the unique dynamics of contact-dependent killing of non-immune, neighboring bacteria, and its potential for significant clinical, ecological, and evolutionary relevance, we propose to call this phenomenon proximicide. […] Proximicide would include, but be broader than, both T6SS-mediated killing and the previously described contact- dependent inhibition (CDI), which refers to a particular two-partner secretion system (CdiB/CdiA) [Aoki et al., 2005; Aoki et al., 2010].

6) The assumption that "predation can occur only at the perimeter of micro-colonies" needs to be supported by some images/data. It would also be helpful to compare model fits to data with and without this assumption.

It has been established by others that T6SS-mediated lysis requires cell-to-cell contact between predator and prey, which can only occur at the boundaries between micro-colonies of predator and prey. The case when all micro-colonies are actually single cells is described in equation 4 (all cells are boundary cells), and we account for the transition between single cells and bona fide micro-colonies in equation 5 (Figure 3—figure supplement 5). We have added citations to support this claim.

7) Section 4.1 ninth paragraph reports adding DNAse to block HGT, but experimental data needs to be supplied (reviewers failed to find them anywhere).

The new experimental data in Figure 6 (to test model predictions) shows HGT inhibition by a dilution series of DNase, and we have updated this section to reference those results, which are more informative than the previously described experiment in which HGT was below detection.

8) The manuscript should extend the description of previous work on fratricide in Streptococcus and competence-mediated neighbor predation and transformation (see comments from reviewer #3).The broader importance of the work should also be discussed (see comments from reviewer #4).

These are good points; we have expanded Discussion as recommended below.

9) The reviewers had two concerns about the title: first, this paper is about the microbiome only in a very minimal sense. Second, the focus in the title on multi-drug resistance suggests a clinical relevance that reviewers felt was lacking, given the focus A. baylyi (a soil organism) and no good reasons for thinking the results would be directly relevant for A. baumanii. A revised title was felt to be appropriate.

To remove “microbiome” and “multi-drug”, we suggest “Microbial Population Dynamics Enhance Horizontal Gene Transfer And Spread Of Antibiotic Resistance”.

Reviewer #2:1) Section 4.1 first paragraph. I don't understand why that first result is 'unexpected'. Is it because the plasmid doesn't have an origin or replication so should not be stably maintained in Acinetobacter, or because the time would be too short to expect transfer? Could they explain this.

We have changed this to “serendipitously”, which may be more accurate. The observation of HGT of GFP to *Acinetobacter* was unexpected because, to our knowledge, functional HGT via natural competence has not previously been observed at single-cell resolution in a microscopic time course experiment, and it was not what we were looking for.

2) In Video 1–Video 6 it is really hard to tell which bacteria is E. coli and which is Acinetobacter. The background fluorescence of the green cells at the start of the video (presumably these are E. coli, since they subsequently lyse and Acinetobacter should not be green yet) is about as high as the 'positive' green at the end of the video in Acinetobacter, so the lacI repression doesn't seem very strong. The authors say that at the end the bacteria are red and green, but in the videos I couldn't see any red + green bacteria – the red all looks bleached so the bacteria are only green. How do we know that these are E. coli? Would it be possible to trace the lineage back through the movie (only for a few, as an example) to show that these green Acinetobacter micro colonies really started as red cells, and better still that they were physically located close to an E. coli cell that subsequently lysed?

We agree it is unfortunate both that the GFP expression in *E. coli* is poorly repressed early in the videos, and that the mCherry expression in *Acinetobacter* becomes bleached later in the movies. We believe both of these phenomena are due to nutrient restriction, which is caused by *Acinetobacter* sticking to and clogging the channels meant to supply nutrients. Indeed, the stickiness of *Acinetobacter* makes microfluidic experiments much more challenging than with only lab *E. coli* strains, and obtaining these videos took many attempts. (Causes of failure, in addition to the usual challenges of microfluidics, included failure to load sufficient numbers of each species in the same trap – *Acinetobacter* adheres to the glass and PDMS within the traps, while *E. coli* does not, washout of less-sticky *E. coli* when the flow rate was raised to reduce clogging by more-sticky *Acinetobacter*, and insufficient time before clogging.)

We have added additional discussion of cell fluorescence in Supplemental note 1.

3) Section 4.1 ninth paragraph, they mention adding DNAse to block HGT – where is this data? I couldn't find it. This is a good control.

The new experimental data in Figure 6 (to test model predictions) shows HGT inhibition by a dilution series of DNase, and we have updated this section to reference those results, which are more informative than the previously described experiment in which HGT was below detection.

4) From the modelling part of the paper they come up with some testable predictions about the role of ratio of predator/prey and the effects of DNAse or competing DNA. Given that they have a really straightforward quantitative assay in their hands, I am wondering why they did not experimentally test some of these hypotheses. I understand the microfluidics is technically challenging, but presumably it would be straightforward enough to repeat their drug resistance acquisition experiments from Figure 2 using a few of the predicted variables?

This is an excellent suggestion. We have performed these experiments and the results are shown in Figure 6 and discussed in the subsection “4.3 Exploration of Environmental Impacts on Horizontal Gene Transfer” and the.

5) Intuition behind many of the modelling results is discussed, but I think it could be worth doing this a bit more thoroughly in the appendix. Often the most useful thing about models is helping to develop this intuition.

This is a good point. We have attempted to discuss the rationale behind the model in the Materials and methods, and a more thorough interpretation of the modeling results in the Discussion.

Reviewer #3:[…] While the model is very interesting and the study nicely contributes to our basic knowledge on HGT, the manuscript tries to sell this in the context of the microbiome (microbiota?), biofilm, and the spread of antibiotics resistance in hospital-acquired pathogens (see title). While all of these areas are extremely important, this study does not address any of these areas. Indeed, A. baylyi is a soil organism and neither a part of the microbiota nor a problem in hospitals (while A. baumannii is, but this organism was not studied here). Thus, sentences like "Therefore, expression of GFP in Acinetobacter should indicate a newly kanamycin-resistant strain, whose de novo appearance demonstrates the clinical relevance of HGT within a microbiome community" seem totally out of place. In general, using key words such a multi-drug resistance (MDR) strains over and over throughout the manuscript seems highly inappropriate. Indeed, all that was shown here are in vitro selected strains in which a single antibiotics gene was transferred by the uptake of a plasmid. This method is commonly used in the transformation field and nobody ever considers such strains as "MDR". The provided experimental data would likewise work if the transfer of a metabolic gene was scored in auxotrophic mutants instead of the transfer of a resistance genes (which is, in fact, only a tool to enumerate transfer efficiencies of any single gene).Another problem is that the title of the manuscript implies information on population dynamics of the microbiome (=microbiota is probably meant). However, the work describes the transfer of plasmids between a laboratory E. coli strain and the soli bacterium A. baylyi. Thus, the stretch to the microbiota is far-fetched. Lastly, the work also does not address bona fide biofilms, which is also highly misleading.In summary, the model part of the study is very interesting and provides novel information on the dynamics of HGT linked to neighbor predation while the experimental part is of limited novelty and lacks methodological information (an extensive list is provided below).

See above response to Essential revisions 1.

1) Acinetobacter baumannii is indeed an opportunistic pathogen and part of the "ESKAPE" pathogens and therefore listed under Centers for Disease Control and Prevention, 2013, (and in a recent report by the WHO) as major threat because of its high level of antibiotic resistance. However, Acinetobacter baylyi, which was used in the experimental settings of the current study, is primarily a soil bacterium with very few studies that indicate that it could also be a causative agent of hospital-acquired infections. Thus, stressing that antibiotic resistance is such a problem in hospital settings and that multi-drug resistance rates exceed 60% in Acinetobacter is too generalized and misleading. Indeed, this is primarily the case for A. baumannii, which is also solely referred to in Antunes, Visca and Towner, 2014. Consequently, the authors should explain to the reader why the experiments were done in A. baylyi instead and they should provide valid arguments why this might anyway be applicable to hospital-acquired pathogens and especially A. baumannii. Using a different non-pathogenic species of the same genus and then concluding that this is also relevant for the pathogenic strains seems inappropriate (e.g., nobody would solely rely on data on the non-pathogenic bacterium Mycobacterium smegmatis to conclude anything of significant importance about the pathogen Mycobacterium tuberculosis). Thus, the authors should provide comparable data for A. baumannii or clearly state that A. baylyi was used as a surrogate for pathogenic Acinetobacter. In the latter case, the manuscript needs to be significantly toned down – especially the Abstract – with respect to the role in antibiotic transmissions in hospitals etc. Indeed, at the start of the last paragraph of the Introduction, "In this work, we extend the observation of killing-enhanced HGT to the clinically threatening Acinetobacter genus" is inappropriate, as most species within the genus Acinetobacter are not at all clinically threatening but soil inhabitants.

See point (i) above.

2) Title "Microbiome Population Dynamics" and Results: "enabling visualization of model microbiomes" – this is irrelevant here, as E. coli alone is not representative of "microbiomes" (even though the E. coli strain was probably a descendant of the original E. coli K12 strain, which was indeed isolated from a stool sample of a patient in 1922) and as far as I am aware of A. baylyi has never described as members of the microbiome because it is in fact a soil organism as stated above (if it is a common member of the microbiota, a reference is for sure needed). Thus, this must be considered as "overselling" and the link with the microbiome is non-existent (except if every E. coli study would be likewise considered a microbiome study, in which case old conjugation papers on F factor would have already shown horizontal gene transfer in the microbiome).

See point (ii) above. To remove “microbiome” and “multi-drug”, we suggest “Microbial Population Dynamics Enhance Horizontal Gene Transfer And Speed Spread Of Antibiotic Resistance”.

3) If the study is meant to be meaningful for A. baumannii and its high prevalence of antibiotic resistance genes, the manuscript should contain the information in T6SS regulation in this organism (e.g., Weber et al., 2015, PNAS). This study shows that multidrug-resistant strains of A. baumannii harbor a self-transmissible resistance plasmid that negatively regulates the T6SS. Thus, following HGT of this plasmid the T6SS is inactive and could have significant consequences, which should be discussed. Along the same line, the authors should provide information about T6SS regulation in A. baylyi.

That observation of Weber at al. is fascinating, and we admit our results do not help us understand what advantages that regulation would provide, if any. However, T6SS regulation varies widely even between strains of the same species, so it may not be appropriate to draw too general conclusions at this point. We have added the following to our discussion of T6SS regulation:

“However, T6SS regulation does not always depend on high cell density, and it can be quite complex and varied, even within strains of the same species, likely reflecting the wide range of ecological contexts and functions performed by the T6SS [Weber et al., 2013; Miyata, Bachmann and Pukatzki, 2013; Bernard et al., 2010; Weber et al., 2015]. […] While T6SS regulation in *A. baylyi* has not been extensively characterized with respect to cell density [Weber et al., 2016], it is active and functional in standard in vitro conditions [Weber et al., 2013; Basler, Ho and Mekalanos, 2013], and we observed T6SS-mediated killing of *E. coli* at all times in the (Video 1–Video 6).

4) The manuscript should extend the description on previous work on fratricide in Streptococcus and competence-mediated neighbor predation and transformation in Vibrio (=same T6SS-mediated killing procedure, as addressed in the current study). In fact, the finding in Figure 2 that T6SS enhances HGT and that "elimination of proximicide impaired HGT by about 100-fold" was shown before for V. cholerae for T6SS-mediated neighbor predation and transformation. Notably, several other experiments such as the nuclease control experiment, the testing of T6SS-mutants, the transformation with purified DNA, the homologous-recombination of genomic DNA that contains homolog regions etc. are likewise the same as in the previous study. Thus, the link between neighbor predation and DNA transfer is not new. The quantitative model, on the other hand, is extremely interesting and should represent the central part of the current manuscript. It is also recommended that the authors discuss whether their model would also be applicable to the previously studied kin-discriminating neighbor-predating bacteria (that is, Streptococci and Vibrios).

We have attempted to clarify the previous observations in *Streptococcus* and *V. cholera*. The phenomena observed in *Streptococcus* represent slightly different dynamics, since the killing is mediated by diffusible bacteriocins, thus enabling longer-range interactions, and it is restricted to certain species. Since HGT facilitated by T6SS-mediated killing has only been reported once, we believe it is valuable to corroborate this in a different species, demonstrating its reproducibility and generalizability. In addition, our observation of in trans complementation of HGT by co-cultured, T6SS+ cells has not yet been reported elsewhere.

Reviewer #4:[…] The mathematical models of HGT via CDI are not necessarily as compelling as the experimental data because of the reality gap between even in vivo and in the environment(s). Even where this was partly closed, the math was left somewhat hanging. As an example, the presence of extracellular DNAse effectively reduces the rate of HGT. This was tested in experiment, and performed as a mathematic model, but I don't see a clear connection made in the text between the model and data demonstrating the effect. I similarly didn't see a clear connection between the competing DNA model and an experiment demonstrating the fit or lack of fit to the model. For the conclusions about likely therapeutic strategies, at least some in vitro data should be clearly fit to the model.

This is an excellent suggestion. We have performed these experiments and the results are shown in Figure 6 and discussed in the subsection “4.3 Exploration of Environmental Impacts on Horizontal Gene Transfer” and the Discussion.

Natural interest in MDR aside, this research has broader importance and the discussion should reflect that. How does this tie into the local taxonomic and genomic heterogeneity of natural populations? Soil, healthy microbiome, unhealthy microbiome? What sort of genes are likely to be exchanged (siderophore uptake, AT pairs, adhesins, phage resistance)? Clearly DNA transfer isn't restricted to a single species; yet homology was demonstrated as accelerating the integration of material into the genome. Some commentary on this is appropriate.

We do indeed expect that neighbor killing-enhanced HGT likely plays a larger and broader role in microbial evolution than currently recognized. Many of these questions are beyond the scope of this paper, but by presenting this data and accompanying model, we hope to both raise awareness of its potential importance, and to provide mathematical tools that will help to study it more quantitatively. We have added some discussion of the broader ecological implications for open questions about microbial evolution with references at the end of the Discussion:

“In addition to helping to explain, predict, and combat the rapid spread of multi drug-resistance, particularly among *Acinetobacter*, the population dynamics revealed here are likely important in the microbiome and microbial evolution more broadly. […] The method presented here for quantifying natural competence with standardized parameters may also help researchers ad- dress other outstanding questions about microbial evolution [Vos et al., 2015].